# Lean Clients, Full Accuracy: Hybrid Zeroth- and First-Order Split Federated Learning

## Abstract

Split Federated Learning (SFL) enables collaborative training between resource-constrained edge devices and a compute-rich server by partitioning deep neural networks. Communication overhead is a central issue in SFL and is well mitigated with auxiliary networks; yet the core client-side computation challenge remains, as back-propagation requires substantial memory and computation costs, severely limiting the scale of models that edge devices can support. To make the client side more resource-efficient, we propose HERON-SFL, a novel hybrid optimization framework that integrates zeroth-order (ZO) optimization for local client training while retaining first-order (FO) optimization on the server. With the assistance of auxiliary networks, ZO updates enable clients to approximate local gradients using perturbed forward-only evaluations per step, eliminating memory-intensive activation caching and avoiding explicit gradient computation in the traditional training process. Leveraging the low effective rank assumption, we theoretically prove that HERON-SFL's convergence rate is independent of model dimensionality, addressing a key scalability concern common to ZO algorithms. Empirically, on ResNet training and large language model (LLM) fine-tuning tasks, HERON-SFL matches benchmark accuracy while reducing client peak memory by up to 64% and client-side compute cost by up to 33% per step, substantially expanding the range of models that can be trained or adapted on resource-limited devices.

## 1 Introduction

Split Federated Learning (SFL) (Thapa et al., 2022; 2021) targets scenarios with resource-constrained clients and compute-rich servers. Under the SFL framework, the full network is cut into client-side and server-side sub-models: each client runs a forward pass up to the cut layer and uploads the intermediate activations; the main server completes the forward pass, computes the loss, back-propagates to the cut layer, and returns the gradients so the client can update its sub-model. In parallel, the federated server (Fed Server) periodically aggregates the clients' weight updates in a federated way, enabling large-scale training that exploits cloud compute while keeping all raw data on-device. However, the update lock (Belilovsky et al., 2020; 2019) imposed by back-propagation means that, at every iteration, each client must idle until the server finishes its backward pass and transmits the cut-layer gradients. This synchronization bottleneck both throttles overall training throughput and amplifies communication overhead in SFL.

To mitigate this bottleneck, recent work equips each client with an auxiliary network (typically a lightweight output layer) that estimates the cut-layer gradients locally (Mu & Shen, 2025; Han et al., 2021; Oh et al., 2022). This design decouples the client from the server, allowing the client sub-model to update immediately without waiting for the server's backward pass, thereby drastically reducing communication overhead and granting extra degrees of freedom for client-side optimization. Previous works show that auxiliary-network SFL not only cuts communication volume by a wide margin but also achieves higher convergence accuracy than the traditional methods (Mu & Shen, 2025; Nair et al., 2025). However, as current approaches take advantage of the auxiliary module as a communication shortcut, they often overlook the significant computational and storage burden it imposes on edge devices. This overhead is primarily driven by the conventional first-order (FO) optimization process, where backpropagation and gradient computation impose prohibitive compute and memory demands on edge devices.

Zeroth-order (ZO) optimization provides an appealing alternative. Unlike FO methods, ZO estimates gradients through parameter perturbations and forward-only evaluations, bringing the computational and storage overhead to a level comparable to those of inference (Malladi et al., 2023). This property suggests that ZO could substantially reduce the client-side burden in SFL. However, ZO optimization is known to suffer from biased gradient estimates and slower convergence compared to FO methods (Qiu et al., 2023), thereby raising an open question:

> *Can ZO methods be effectively integrated into SFL to reduce client-side computation and storage without sacrificing accuracy or convergence guarantees?*

We answer this question affirmatively by proposing HERON-SFL, a novel Hybrid zEroth- and fiRst-Order optimizatioN framework for SFL. In HERON-SFL, clients replace conventional FO gradient computation with lightweight ZO updates, applied to the local model (comprising the client-side and auxiliary networks). This design eliminates the need for backpropagation and caching, enabling edge devices to operate with markedly reduced memory and compute budgets. Importantly, clients transmit only the smashed activations required for the server-side FO training, while the server performs standard updates on its own partition of the model.

Our main contributions are summarized as follows:

- We propose HERON-SFL, a novel hybrid zeroth- and first-order SFL framework. Building upon an auxiliary network that enables decoupled local training, we introduce zeroth-order (ZO) optimization on the client side. This eliminates the need for backpropagation for local updates, thereby significantly reducing on-device memory and computational costs.

- We provide the first theoretical study of hybrid ZO–FO optimization in SFL. Our analysis shows that under a low effective rank assumption, HERON-SFL achieves an $\mathcal{O}(1/\sqrt{T})$ convergence rate, which matches that of standard FO approaches. This result shows that the usual ZO slowdown can be alleviated under the proposed hybrid structure and assumptions, yielding a convergence rate comparable to FO methods.

- We conduct comprehensive experiments spanning both vision (ResNet training) and language (LLM fine-tuning) tasks. Results show that HERON-SFL consistently reduces client peak memory by up to 64% and client computation per step by up to 33%, while matching the accuracy of state-of-the-art, auxiliary-network-based FO SFL methods. These gains highlight HERON-SFL's practical potential for deploying advanced models on previously infeasible devices.

## 2 RELATED WORK

**Split Federated Learning**. While modern foundation models achieve state-of-the-art performance (Brown et al., 2020; Chowdhery et al., 2023), their immense computational and memory requirements restrict them to data centers, limiting their real-world reach (Luo et al., 2024; Sani et al., 2024). To bring large language models onto edge devices, SFL was proposed by merging Federated Learning (FL) (McMahan et al., 2017) with Split Learning (SL) (Vepakomma et al., 2018) to enhance data privacy and robustness (Thapa et al., 2022; Lee et al., 2024). However, SFL remains constrained by the SL training paradigm, leading to prohibitive communication overhead and a synchronous update lock, as clients must await gradients from the server before updating (Kairouz et al., 2021; Vepakomma et al., 2018). To mitigate these bottlenecks, recent research has primarily pursued two complementary directions: system-level optimization and algorithmic decoupling.

*System-level optimization* aims to adapt the SFL protocol to the constraints of edge networks. This includes methods for adaptive model splitting based on network conditions (Lin et al., 2024b), hierarchical topologies to manage client resources (Lin et al., 2025), parallel training designs optimized for wireless networks (Wu et al., 2023), and dynamic resource-based tiers to speed up FL/SFL training under heterogeneous environments Mohammadabadi et al. (2024). Underpinning these practical advances, recent theoretical work has also focused on providing formal convergence guarantees for SFL, particularly under realistic conditions such as data heterogeneity (Han et al., 2024; Li & Lyu, 2023).

*Algorithmic decoupling* aims to eliminate the synchronous lock by incorporating a client-side auxiliary model to decouple client and server updates by generating local gradient estimates, thereby obviating the need for server-to-client gradient transmission (Han et al., 2021; Mu & Shen, 2025;

Oh et al., 2022; Nair et al., 2025). Inspired by decoupled training (Belilovsky et al., 2020; 2019) in centralized settings, this strategy can nearly halve communication costs. Despite their demonstrated efficacy, these auxiliary models introduce a significant trade-off: a substantial increase in the client's computational and memory footprint, as the auxiliary network can be considerably larger than the primary client-side model itself (Nair et al., 2025). Group Knowledge Transfer (He et al., 2020) is another relevant approach that transfers logits from client-side auxiliary models to the server via knowledge distillation (Hinton et al., 2015), although it differs from SFL in formulation and training objective.

**Zeroth-Order Optimization for Distributed Machine Learning**. ZO optimization estimates gradients through function evaluations (Liu et al., 2020), making it particularly useful when explicit gradients are unavailable, such as in reinforcement learning (Nakashima & Kobayashi, 2025; Lei et al., 2022; Zhang & Ying, 2024) and privacy-sensitive scenarios (Chen et al., 2017; Liu et al., 2018; 2019). Recently, ZO has gained attention as an efficient strategy for training (Chen et al., 2024) and fine-tuning (Malladi et al., 2023), since it avoids back-propagation's memory and compute overhead. In distributed machine learning, ZO has been explored as a gradient estimator in FL, demonstrating promising benefits in privacy preservation (Zhang et al., 2021; Fang et al., 2022; Ling et al., 2024) and communication reduction (Li et al., 2024). However, its adoption in the SFL framework remains limited. The main barrier is that variance reduction in ZO requires multiple perturbations, which would substantially increase intermediate activation transmissions and thus communication overhead. To address this, we *restrict ZO to the client side* with the help of auxiliary networks, enabling resource-efficient training while avoiding additional communication costs.

## 3 ALGORITHM DESIGN

### 3.1 SFL WITH AUXILIARY NETWORK

We consider an SFL system with one server and $N$ clients, each holding a private dataset $\mathcal{D}_i$, where the entire dataset is the set $\{\mathcal{D}_i\}_{i=1}^N$. The global model is split at a cut layer into client- and server-side sub-models, where we denote the collection of parameters as $\boldsymbol{\theta}_g = \{\boldsymbol{\theta}_c, \boldsymbol{\theta}_s\}$. Each client $i$ owns a local version of the client-side model, $\boldsymbol{\theta}_{c,i}$. For a sample $\xi_{i,j} \in \mathcal{D}_i$, client $i$ performs a forward pass up to the cut layer to produce the smashed data, $\boldsymbol{s}_i = \boldsymbol{\theta}_{c,i}(\xi_{i,j})$, and uploads it to the main server. The server then completes the forward pass by processing these activations with its sub-model $\boldsymbol{\theta}_s$. The goal is to minimize the global loss function:

$$\min_{\boldsymbol{\theta}_c, \boldsymbol{\theta}_s} \ f(\boldsymbol{\theta}_g) = \frac{1}{N} \sum\nolimits_{i=1}^N f_i(\boldsymbol{\theta}_g) = \frac{1}{N} \sum\nolimits_{i=1}^N \mathbb{E}_{\xi_{i,j} \sim \mathcal{D}_i}[\ell(\boldsymbol{\theta}_g; \xi_{i,j})], \tag{1}$$

where $f_i(\boldsymbol{\theta}_g)$ and $f(\boldsymbol{\theta}_g)$ measure the expected loss on the global model over client $i$'s local dataset $\mathcal{D}_i$ and the entire dataset, respectively, computed using a task-specific loss function $\ell(\cdot)$ (e.g., cross-entropy).

We adopt the SFLV2 style framework: a single server-side model $\boldsymbol{\theta}_s$ resides on the main server and is trained by sequentially processing smashed data $\boldsymbol{s}_i$ from all clients, while a Fed-Server aggregates client-side parameters into the average $\bar{\boldsymbol{\theta}}_c := \frac{1}{N} \sum_i \boldsymbol{\theta}_{c,i}$ (initial parameters for the next round). To reduce communication overhead and enable client-side local feedback, each client $i$ attaches an auxiliary (Aux) model $\boldsymbol{\theta}_{a,i}$ to form a local predictor $\boldsymbol{\theta}_{l,i}(\xi_{i,j}) = \boldsymbol{\theta}_{a,i}(\boldsymbol{\theta}_{c,i}(\xi_{i,j}))$, where $\boldsymbol{\theta}_{l,i} = \{\boldsymbol{\theta}_{c,i}, \boldsymbol{\theta}_{a,i}\}$ (Mu & Shen, 2025; Oh et al., 2022). Because of the Aux model, the SFL system breaks the *training lock* between $\boldsymbol{\theta}_c$ and $\boldsymbol{\theta}_s$: by leveraging $\boldsymbol{\theta}_a$, the client can perform local updates independently of server-side gradient feedback.

After initializing the global model $\{\boldsymbol{\theta}_c, \boldsymbol{\theta}_s\}$, the basic SFL-Aux algorithm proceeds: in each round, client $i$ computes smashed data $\mathcal{S}_i = \boldsymbol{\theta}_{c,i}(\xi_i)$ on local mini-batches $\xi_i = \{\xi_{i,j}\}_{j=1}^B$ and uploads them to the Main-Server, while updating $\boldsymbol{\theta}_{l,i}$ by minimizing a local loss from $\boldsymbol{\theta}_{a,i}(\mathcal{S}_i)$, with backpropagation confined to the client. The Main-Server queues smashed data from all clients and sequentially executes forward/backward passes to update $\boldsymbol{\theta}_s$. After a fixed number of local steps, the Fed-Server aggregates all participated $\boldsymbol{\theta}_{l,i}$ (e.g., via weighted averaging like FedAvg (McMahan et al., 2017)) and broadcasts updated global model $\bar{\boldsymbol{\theta}}_l$ to all clients to initiate the next round.

## 3.2 ZEROTH-ORDER GRADIENT ESTIMATOR

Unlike prior methods that rely on full forward and backward passes through the client and its aux­iliary network to compute first-order gradients $\nabla\ell(\boldsymbol{\theta}_l; \xi_i)$, we adopt a mini-batch-type stochastic gradient estimator with two-point evaluation. Specifically, for function $f_{l,i}$, the two-point type stochastic ZO gradient estimator is defined as:

$$\hat{\nabla} f_{l,i}(\boldsymbol{\theta}_l; \xi_i)) = \frac{1}{B} \sum_{j=1}^{B} \frac{d\boldsymbol{u}}{\mu} [\ell_{l,i}(\boldsymbol{\theta}_l + \mu\boldsymbol{u}; \xi_{i,j}) - \ell_{l,i}(\boldsymbol{\theta}_l; \xi_{i,j})], \tag{2}$$

where $\boldsymbol{u}$ is a random vector drawn from either a Gaussian or a Uniform ball distribution, $\mu$ is a positive perturbation step size. This estimator approximates the smoothed objective function's gradient. Formally, it can be shown that this estimator is an unbiased estimate of $\nabla f_{l,i}^{\mu}(\boldsymbol{\theta}_l)$, where $f_{l,i}^{\mu}$ is the Gaussian-smoothed surrogate of the original function $f_{l,i}$. The bias with respect to the true gradient $\nabla f_{l,i}$ is therefore introduced by the smoothing process itself and is controlled by the parameter $\mu$. We defer the formal definition of the smoothed function and its properties to Appendix A.2.

## 3.3 PROPOSED ALGORITHM

We now summarize the end-to-end training pro­cess of our proposed framework, which op­erates over a series of communication rounds (high-level illustration depicted in Figure 1). Each round, indexed by $t$, encompasses four key stages: model initialization, local client computation, server-side updates, and local model aggregation in Fed Server. The entire process is formalized as follows:

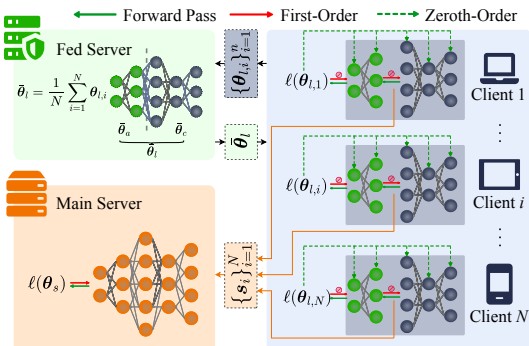

Figure 1: The proposed HERON-SFL algorithm.

**1. Model Initialization.** At the start of the $t$-th communication round, the Fed-Server broad­casts the global model parameters $\boldsymbol{\theta}_c^t$ and $\boldsymbol{\theta}_a^t$ that are resulted from the federated aggregation at the end of last round. Upon receiving these parameters, each client $i$ initializes its local models for the subsequent update process: $\boldsymbol{\theta}_{l,i}^{t,0} = \{\boldsymbol{\theta}_{c,i}^{t,0}, \boldsymbol{\theta}_{a,i}^{t,0}\} = \{\boldsymbol{\theta}_c^t, \boldsymbol{\theta}_a^t\}$.

**2. Local Model Update and Smashed Data Upload.** The client then proceeds with $h$ local model updates. During this process, the update of the client-side model is decoupled from the server-side model by leveraging an auxiliary network. Distinct from existing methods, our paradigm employs a ZO gradient estimator (defined in Eq. 2) to approximate the gradients of a local loss function. This allows the client to perform timely updates without requiring traditional back-propagation from the server. After performing $h$ local gradient descent steps, the cumulative update for the client-side models can be concisely written as:

$$\boldsymbol{\theta}_{l,i}^{t,h} = \boldsymbol{\theta}_{l,i}^{t,0} - \eta_l \sum_{m=1}^{h} \hat{\nabla} f_{l,i}(\boldsymbol{\theta}_{l,i}^{t,m}; \xi_i)) \tag{3}$$

During the local update phase, the client uploads its smashed data to the server every $k$ local steps for the subsequent server-side training phase.

**3. Server Model Update.** The server receives the smashed data from each client $i$ and performs model updates sequentially using an SFLV2 (Thapa et al., 2022) training scheme. In this setting, each client's smashed data is processed one-by-one, and standard first-order optimization based on forward and backward propagation is used to estimate gradients and update the server-side model parameters $\boldsymbol{\theta}_s^t$ accordingly:

$$\boldsymbol{\theta}_s^{t+1} = \boldsymbol{\theta}_s^t - \eta_s \sum_{i=1}^{N} \frac{1}{|\mathcal{D}_i|} \sum_{\xi_i \in \mathcal{D}_i} \nabla\ell(\boldsymbol{\theta}_s^t; \boldsymbol{\theta}_{c,i}^t(\xi_i)), \tag{4}$$

where $\nabla_{\boldsymbol{\theta}_s} l(\boldsymbol{\theta}_s^t; \boldsymbol{\theta}_{c,i}^t(\xi_i))$ is the real gradient of the server-side loss function using back propagation.

**4. Model Aggregation in Fed Server.** Upon completion of the $h$ local updates, each client transmits its updated local parameters $\boldsymbol{\theta}_{l,i}^{t,h}$ to the fed server for aggregation. The fed server averages these parameters across all $N$ clients to compute the global model combined by client-side and auxiliary models for the next round:

$$\boldsymbol{\theta}_l^{t+1} = \bar{\boldsymbol{\theta}}_l^t = \frac{1}{N} \sum_{i=1}^{N} \boldsymbol{\theta}_{l,i}^{t,h} \tag{5}$$

The server-side model, $\boldsymbol{\theta}_s^{t+1}$, which was updated sequentially during the round, is already finalized and requires no aggregation. Finally, the new global model $\boldsymbol{\theta}_g^{t+1} = \{\boldsymbol{\theta}_c^{t+1}, \boldsymbol{\theta}_s^{t+1}\}$ is assembled and prepared for distribution in the subsequent communication round.

In essence, HERON-SFL replaces the clients' local updates in standard SFL with updates driven by a ZO gradient estimator, while retaining client-side auxiliary networks to guide local learning. This design eliminates the need for explicit backpropagation on resource-constrained devices: clients only perform a small number of forward computations and randomized probes to update parameters, substantially reducing compute and memory demands. Clients periodically upload smashed data (every $h$ local steps) to supply the server with the activations required for its independent FO training on the server-side model. A critical concern, however, is that ZO optimization is often associated with slow convergence. In the following sections, we will demonstrate both theoretically and empirically that HERON-SFL overcomes this potential challenge within the SFL framework.

# 4 CONVERGENCE AND RESOURCE CONSUMPTION ANALYSIS

## 4.1 CONVERGENCE ANALYSIS

In this section, we provide a formal convergence analysis to establish the theoretical guarantees for the proposed FSL-HERON framework. For the sake of clarity and conciseness, the detailed mathematical proofs are deferred to Appendix A. The theoretical framework is built upon the following standard assumptions, which are widely adopted in the analysis of distributed optimization algorithms (Karimireddy et al., 2020; Reddi et al., 2020; Mu & Shen, 2025; Fang et al., 2022).

**Assumption 4.1 (L-smoothness).** *The loss functions of clients and server are L-smooth. Mathematically, for any $\boldsymbol{x} \in \mathbb{R}^d$ and $\boldsymbol{y} \in \mathbb{R}^d$, the following holds:*

$$\|\nabla f(\boldsymbol{x}) - \nabla f(\boldsymbol{y})\| \leq L\|\boldsymbol{x} - \boldsymbol{y}\|, \quad f(\boldsymbol{y}) \leq f(\boldsymbol{x}) + \nabla f(\boldsymbol{x})^T (\boldsymbol{y} - \boldsymbol{x}) + \frac{L}{2}\|\boldsymbol{y} - \boldsymbol{x}\|^2, \tag{6}$$

*where $f$ is the loss function, and $L$ is the Lipschitz constant.*

**Assumption 4.2 (Bounded gradients).** *The gradients of the local loss function $\ell_i(\boldsymbol{\theta}_c, \boldsymbol{\theta}_s)$ are bounded, i.e., there exists a constant $G$ such that:*

$$\|\nabla_{\boldsymbol{\theta}_c} \ell_i(\boldsymbol{\theta}_c)\|^2 \leq G_c^2, \|\nabla_{\boldsymbol{\theta}_s} \ell_i(\boldsymbol{\theta}_s)\|^2 \leq G_s^2. \tag{7}$$

**Assumption 4.3 (Bounded variance).** *The variance of the zeroth-order gradient estimator is bounded, i.e., there exists a constant $\sigma^2$ such that:*

$$\mathbb{E}[\|\hat{\boldsymbol{g}}_{c,i}^{t,m} - \nabla_{\boldsymbol{\theta}_c} f_i(\boldsymbol{\theta}_c, \boldsymbol{\theta}_s)\|^2] \leq \sigma^2. \tag{8}$$

**Assumption 4.4 (Convergence of client sub-model).** *For each client $i$ at global round $t$, let $z_{c,i}^t = g_{x_{c,i}^t, h}(z)$ be the output of the $i$-th client-side model (with input determined by $x_{c,i}^t$ and $\mathcal{D}_i$), and denote by $P_{c,i}^t(z)$ its output distribution. Let $P_{c,i}^*(z)$ be the reference (time-invariant) output distribution of the $i$-th client-side model evaluated at $x_c^*$ and $\mathcal{D}_i$. Define the distribution distance*

$$d_{c,i}^t := \int_{\mathcal{Z}} \big| P_{c,i}^t(z) - P_{c,i}^*(z) \big| \, dz, \tag{9}$$

*i.e. the $L_1$ (total-variation) distance between $P_{c,i}^t$ and $P_{c,i}^*$. We assume that the aggregate drift across clients is uniformly bounded as follows:*

$$\frac{1}{T} \sum_{t=1}^{T} \sum_{i=1}^{N} d_{c,i}^t \leq \delta, \text{ and } \delta < \infty. \tag{10}$$

**Remark 1** Together, the Assumptions above ensure a well-behaved optimization environment. Assumption 4.1 guarantees Lipschitz-continuous gradients and provides the usual quadratic upper bound used in descent arguments; Assumption 4.2 prevents arbitrarily large client/server updates and thus promotes numerical stability; and Assumption 4.3 limits the stochastic error between the estimator and the true gradient. Assumption 4.4 is tailored to the auxiliary-network-assisted FSL setting, as also adopted in Mu & Shen (2025) and motivated by centralized synthetic-gradient frameworks (Belilovsky et al., 2020). This condition is essential for guaranteeing the stability and convergence of the SFL process under local gradient updates.

**Theorem 4.5** (**Convergence rate of HERON-SFL in i.i.d. setting**). *Under Assumptions 4.1–4.4 , if the client learning rate satisfies $\eta_c \leq \{ \frac{1}{3Lh}, \frac{2}{NLh^2}, \frac{N}{72L} \}$, and is chosen as $\eta_c = \mathcal{O}(\sqrt{(NB)/(dhT)})$ while the server learning rate is set to $\eta_s = \mathcal{O}(\sqrt{(hB)/(dNT)})$, and perturbation step size is set to $\mu = \mathcal{O}(1/(dhNBT)^{1/4})$. The convergence rate of the HERON-SFl algorithm can be guaranteed as:*

$$\min_{t \in [T]} \mathbb{E} \left[ \|\nabla f(\boldsymbol{\theta}_g^t)\|^2 \right] \leq \mathcal{O} \left( \sqrt{\frac{d}{hNBT}} \right) + \mathcal{O} \left( \sqrt{\frac{1}{dhNBT}} \right). \tag{11}$$

**Remark 2** The derived bounds on the expected gradient norm indicate that the algorithm can achieve a favorable trade-off between the model complexity (characterized by the dimensionality $d$) and the training batchsize (captured by $B$) over the training horizon $T$. The bound is dominated by $\mathcal{O}(\sqrt{d/(hNBT)})$ (the second term is smaller by $1/\sqrt{d}$). Thus, larger $N$ or $B$ linearly reduces the required rounds; increasing local steps $h$ improves the rate as $1/\sqrt{h}$, trading fewer communication rounds for more local computation. The dependence on model size is $\sqrt{d}$ (or $d$ in sample complexity), which is the drawback of ZO optimization: convergence degrades with increasing dimensionality. Below, we show that the dependency on $d$ can be reduced under structural assumptions on an effective dimension.

**Assumption 4.6** (**Low $\kappa$-Effective Rank**). *Let $G_t \triangleq \max_{i, \xi_i \subset \mathcal{D}_i} \|\nabla_{\boldsymbol{\theta}_l} l_l(\theta_{l,i}^t; \xi_i)\|$. There exists a Hessian matrix $H_l(\theta_{l,i}^t) \preceq L \cdot I_{d_l}$ such that:*

- *For all $\boldsymbol{\theta}_l$ such that $\|\boldsymbol{\theta}_l - \theta_{l,i}^t\| \leq 2\eta_c d_l G_t$, we have $\nabla^2 l_l(\boldsymbol{\theta}_l) \preceq H_l(\theta_{l,i}^t)$.*
- *The effective rank of $H_l(\theta_{l,i}^t)$, i.e., $\frac{tr(H_l(\theta_{l,i}^t))}{\|H_l(\theta_{l,i}^t)\|_2}$, is at most $\kappa$.*

**Theorem 4.7** (**Convergence Rate of HERON-SFL with Low Effective Rank Assumption**). *Under Assumptions 4.1–4.6 ,if the client learning rate satisfies $\eta_c \leq \frac{1}{4L}(1 + \frac{d\kappa + d - 2}{d+2})$ and $\mu \leq \frac{\sqrt[4]{\kappa}}{\sqrt[4]{NT} \sqrt{(d+3)^3}}$, and is chosen as $\eta_c = \mathcal{O}(\sqrt{(NB)/(\kappa T)})$ while the server learning rate is set to $\eta_s = \mathcal{O}(\sqrt{B/(\kappa NT)})$. The convergence rate of the HERON-SFL algorithm can be guaranteed as:*

$$\min_{t \in [T]} \mathbb{E} \left[ \|\nabla f(\boldsymbol{\theta}_g^t)\|^2 \right] \leq \mathcal{O} \left( \sqrt{\frac{\kappa}{NBT}} \right) + \mathcal{O} \left( \frac{1}{T} \right) + \frac{2}{\delta} \left[ \frac{2G_s^2}{N(2N-1)} \Delta + \frac{\mu^2 L^2}{2}(d+3)^3 \right]. \tag{12}$$

**Remark 3** With the prescribed $\mu$, the smoothing bias term $\propto \mu^2(d+3)^3$ is at most $\mathcal{O}(\sqrt{\kappa/(NT)})$ and the drift term vanishes in the i.i.d. case ($\Delta = 0$), so the bound simplifies to $\mathcal{O}(\sqrt{\kappa/(NBT)}) + \mathcal{O}(1/T)$, which is independent with the model dimension $d$, removing the usual $\sqrt{d}$ degradation of ZO methods and matching the $1/\sqrt{T}$ rate of FO SFL (Mu & Shen, 2025; Nair et al., 2025) up to condition number $\kappa$ factors.

## 4.2 Client-side Resource Cost Analysis

The following analysis, summarized in Table 1, compares the per-client resource consumption for a single parameter update step on a fixed-size batch of data, assuming all other hyperparameters are held constant. Let $p$ be the data size of one local batch, $q$ be the size of the smashed layer, and $|\boldsymbol{\theta}_c|$, $|\boldsymbol{\theta}_a|$ be the size of the client-side and auxiliary models, respectively.

**Communication Load.** The primary communication advantage of decoupled frameworks (CSE-FSL, FSL-SAGE, and HERON-SFL) over traditional SFL (SFLV1/V2) stems from the elimination of the server-to-client gradient download. While traditional SFL requires a two-way intermediate

Table 1: Client-Side Resource Costs per Local Update.

| Method | Comms. per Client | Peak Memory | FLOPs |
|---|---|---|---|
| SFLV1/V2 | $2pq + 2|\boldsymbol{\theta}_c|$ | $\mathcal{O}(|\boldsymbol{\theta}_c|)$ | $3F_c$ |
| CSE-FSL / FSL-SAGE | $pq + 2(|\boldsymbol{\theta}_c| + |\boldsymbol{\theta}_a|)$ | $\mathcal{O}(|\boldsymbol{\theta}_c| + |\boldsymbol{\theta}_a|)$ | $3(F_c + F_a)$ |
| HERON-SFL | $pq + 2(|\boldsymbol{\theta}_c| + |\boldsymbol{\theta}_a|)$ | $\mathcal{O}(1)$ | $n_p(F_c + F_a)$ |

data exchange for each batch (represented by the term $2pq$), decoupled methods perform only a one-way upload, halving this cost to $pq$. The trade-off for this gain is the added cost of exchanging the auxiliary model parameters, $|\boldsymbol{\theta}_a|$. Nevertheless, this parameter exchange typically represents a minor communication burden relative to the transmission of smashed data.

**Peak Memory.** FO frameworks like SFLV1/V2 and CSE-FSL require caching intermediate activations for backpropagation. This results in a peak memory footprint that scales with the size of the locally trained models, i.e., $\mathcal{O}(|\boldsymbol{\theta}_c|)$ and $\mathcal{O}(|\boldsymbol{\theta}_c| + |\boldsymbol{\theta}_a|)$ respectively. This overhead can be an order of magnitude larger than that of inference (Griewank & Walther, 2008). In contrast, the ZO-based HERON-SFL obviates activation caching, reducing its peak memory to $\mathcal{O}(1)$, which is equivalent to that of inference (Malladi et al., 2023).

**Remark 4** Local ZO updates are highly memory-efficient for two primary reasons. First, they eliminate the need for backpropagation, thus avoiding the high cost of caching intermediate activations. Second, the perturbed parameters $\boldsymbol{u}$ generated in the calculation $\hat{\nabla} f_{l,i}(\boldsymbol{\theta}_l; \xi_i))$ do not require storing the full perturbation vector; instead, the vector can be procedurally generated from a single random seed and applied in-place, further minimizing memory overhead.

**FLOPs.** Assuming a backward pass is twice as computationally expensive as a forward pass ($F$) (Chen et al., 2016), first-order methods incur a cost of approximately $3F_c$ (for SFLV1/V2) or $3(F_c + F_a)$ (for CSE-FSL and FSL-SAGE) per update, where $F_c$ and $F_a$ are the forward pass costs of the client and auxiliary models, respectively. In contrast, HERON-SFL performs ZO updates directly on the client, similar to the approach in MeZO (Malladi et al., 2023). In practice, a standard two-point ZO estimator is typically sufficient for stable and effective parameter updates, requiring a computational cost of $2(F_c + F_a)$ in HERON-SFL.

## 5 EXPERIMENTS

### 5.1 EXPERIMENT SETTING

In this section, we conduct experiments on both model training and fine-tuning to show the performance of our proposed HERON-SFL algorithm[1]. For comparison, we use the following baseline methods: SFLV1/V2 (Thapa et al., 2022) or SplitLoRA (Lin et al., 2024a)[2], CSE-FSL (Mu & Shen, 2025), and FSL-SAGE (Nair et al., 2025). We conduct the experiments under two complementary training paradigms, implementing all models in PyTorch and running them on NVIDIA RTX A6000 NVL GPU (48 GB):

**Full Training from Scratch**. We study the convergence of ResNet-18 (He et al., 2016) under SFL on CIFAR-10 (Krizhevsky et al., 2009) with 5 clients. The model is split after the second 2-D BatchNorm layer; the client holds the front part while the server holds the back part. An auxiliary head consisting of a single fully connected layer is attached to the cut layer. Unless otherwise stated, we adopt the hyperparameters in Thapa et al. (2022): batch size 256 and Adam optimizers on both sides with a learning rate of $1e{-}4$.

---

[1]Our source code is available at `https://anonymous.4open.science/r/HERON-SFL-BB31/`.

[2]While SFLV1/V2 are designed for the training-from-scratch paradigm, our focus on the distinct task of language fine-tuning led to the development of SplitLoRA, which integrates LoRA with the SFLV2 framework. We omit a comparison with an SFLV1-based approach because its need for multiple server models is computationally prohibitive for large-scale models.

**Language Model Fine-tuning**. We fine-tune GPT2-Small and GPT2-Medium (Radford et al., 2021) on the E2E dataset Novikova et al. (2017) with 3 clients. Unless specified otherwise, for GPT2-Small, the model is split after the third transformer block, with an auxiliary network consisting of one transformer block and the unembedding layer. For GPT2-Medium, the split occurs after the sixth block, with a three-block auxiliary network plus the unembedding layer. As the auxiliary network is not pre-trained, we initialize its parameters by copying the weights from the initial blocks of the server-side model. All components are fine-tuned using Low-Rank Adaptation (LoRA) (Hu et al., 2022), where only adapters of rank 8 are updated and all other parameters are frozen.

The former setting evaluates whether SFL can train a model *from scratch*, a prerequisite when no reliable checkpoint exists. The latter mirrors the prevailing industrial practice of pre-training a large language model once and then adapting it with parameter- and memory-efficient techniques such as LoRA. By examining both regimes, we separately measure the contributions of data-parallel federation, model partitioning, and parameter-efficient adapters, and we show that HERON-SFL consistently outperforms strong baselines in both scenarios.

## 5.2 Training from Scratch: ResNet18 on Cifar-10

**Convergence Behavior.** Figure 2 illustrates the test accuracy of each method versus the number of communication rounds. In the IID setting, our proposed HERON-SFL shows convergence behavior nearly identical to other auxiliary-network baselines like CSE-FSL and FSL-SAGE[3], with all three performing slightly below the top-performing SFLV2. A similar trend is observed in the more challenging non-IID setting, which confirms that our hybrid algorithm achieves convergence comparable to its first-order counterparts.

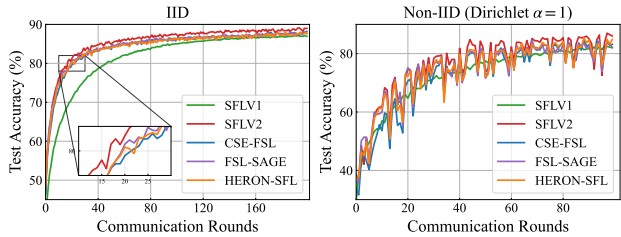

Figure 2: ResNet-18 test accuracy vs. communication rounds on CIFAR-10 for IID (left) and non-IID (right) distributions.

Table 2: Client consumptions for ResNet-18 on CIFAR-10.

| Algorithms | Comm. (GB) | Peak FP (MB) | FLOPS (G) |
|---|---|---|---|
| SFLV1 | 1216.00 | 709.93 | 59.51 |
| SFLV2 | 390.67 | | |
| CSE-FSL | 258.55 | 726.46 | 59.85 |
| FSL-SAGE | 244.24 | | |
| HERON-SFL | **244.19** | **259.44** | **39.90** |

**Communication, Storage, and Computational Costs.** Table 2 provides a quantitative comparison of the resource consumption on the client side. In terms of communication load, HERON-SFL is among the most efficient methods, requiring only 244.19 GB of total communication, a volume nearly identical to FSL-SAGE (244.24 GB) and superior to all other baselines.

The most significant advantages of HERON-SFL are evident in its on-device resource requirements. By eliminating client-side backpropagation, it drastically reduces the peak memory footprint (Peak FP) to just 259.44 MB—a reduction of approximately 63% compared to the almost 710 MB required by SFLV1 and SFLV2. Similarly, the computational cost (FLOPs) is lowered to 39.90 G FLOPs, a reduction of over 33% compared to the ~59 G FLOPs of other methods. This substantial decrease in both storage and compute burden confirms that HERON-SFL is highly suitable for deployment in resource-constrained environments.

## 5.3 Language Model Fine-tuning

For the task of language model fine-tuning, HERON-SFL demonstrates superior communication efficiency and faster convergence. As illustrated in Figure 3, its validation perplexity decreases more

---

[3]We note that FSL-SAGE does not exhibit a significant advantage in our experiments, which we attribute to our design choice of using a minimal auxiliary network purely for decoupling the updates of server and clients. This contrasts with the approach in (Nair et al., 2025), where the alignment mechanism of FSL-SAGE is more impactful as the auxiliary model is intentionally designed to be even larger than the client model, thus requiring explicit alignment to ensure consistency with the server's task.

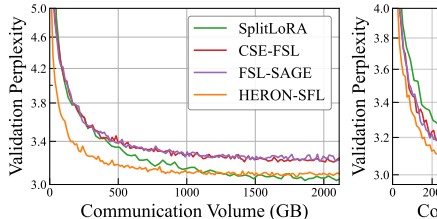

Figure 3: GPT2 perplexity curves vs. Communication Volume on E2E for small (left) and medium (right) models.

Table 3: Client consumptions for GPT2-Medium on E2E.

| Algorithms | Peak FP (GB) | FLOPS (T) |
|---|---|---|
| SplitLora | 4.59 | 5.68 |
| CSE-FSL | 9.09 | 9.48 |
| FSL-SAGE | | |
| HERON-SFL | **4.03** | **5.26** |

rapidly than the baselines for both GPT2-Small and GPT2-Medium. Notably, for GPT2-Small, HERON-SFL converges faster and achieves a final perplexity that is competitive with SplitLoRA while outperforming both CSE-FSL and FSL-SAGE. While all methods reach a similar performance on GPT2-Medium, HERON-SFL does so with significantly less communication costs, and even slightly surpasses CSE-FSL and FSL-SAGE on GPT2-Small. This mild performance gain is consistent with recent findings in ZO-based LLM fine-tuning, where the update landscape exhibits strong low-rank structure, making zeroth-order steps exceptionally effective. Similar behavior is reported in MeZO (Malladi et al., 2023), which shows that ZO fine-tuning can match or even surpass first-order methods under comparable settings.

Echoing the resource efficiency observed in the ResNet experiments, HERON-SFL substantially lowers the on-device computational and memory burden for clients. Table 3 provides a clear comparison of the resource consumption per local update. HERON-SFL requires a peak memory (Peak FP) of only 4.03 GB, which is less than half that of CSE-FSL (9.09 GB) and also more efficient than the SplitLoRA baseline (4.59 GB). The reduction in computational cost is even more pronounced, with HERON-SFL needing only 5.26 TFLOPS, a decrease of approximately 44% compared to CSE-FSL and FSL-SAGE. This reduction in both memory footprint and floating-point operations confirms that by eliminating client-side backpropagation, our method significantly lowers the hardware barrier, making it feasible to fine-tune large language models on resource-constrained edge devices.

## 5.4 ABLATION STUDY OF LOCAL MODEL COMPLEXITY

We investigate the impact of local model complexity on the GPT2-medium fine-tuning task. In this ablation study, we evaluate two primary scenarios: one where the client-side model contains the initial 3 transformer blocks, and another with 6 blocks. For each scenario, we vary the auxiliary network's architecture from a lightweight base (LayerNorm and unembedding layers only) to progressively larger versions containing one, two, or three transformer blocks. Figure 4 plots the final training loss after a fixed number of training rounds. The results show that our proposed HERON-SFL is largely insensitive to the complexity of the auxiliary network; in both the 3-block and 6-block settings, it achieves a strong final training loss even with the simplest auxiliary model. In contrast, the

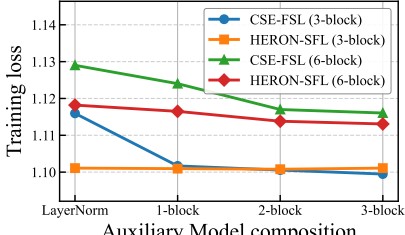

Figure 4: Effect of aux-model complexity.

performance of the first-order baseline, CSE-FSL, is highly dependent on a more powerful auxiliary model, showing a clear trend of improvement as the network becomes more complex. This suggests that for ZO-based methods, there is little justification for using a resource-intensive auxiliary network, whereas first-order methods require one to reach their full potential.

This study validates the comprehensive efficiency of HERON-SFL, which stems from two key advantages. First, its use of zeroth-order optimization reduces the peak memory footprint to the level of inference by eliminating backpropagation. Second, it attains excellent global convergence while requiring only a minimal auxiliary architecture. Crucially, these resource savings do not come at the cost of performance; our experimental results highlight the dual advantages of HERON-SFL in achieving both fast convergence and low on-device costs. This provides a superior performance-to-cost balance when compared to first-order baselines like FSL-SAGE and CSE-FSL.

## 6 CONCLUSION

This work proposes HERON-SFL, a novel hybrid ZO-FO framework that addresses the critical computation and memory limitations on edge devices within SFL. It performs ZO optimization on edge devices to eliminate costly backpropagation, thereby significantly reducing on-device memory and computational requirements. Empirical and theoretical analysis demonstrate that the framework not only achieves a theoretical convergence rate of $\mathcal{O}(1/\sqrt{T})$ independent of model dimensionality under the low effective rank assumption, but also empirically matches the accuracy of SFL benchmarks on diverse tasks while substantially reducing client-side resource costs.

Future work may explore non-differentiable objectives—for example, directly optimizing evaluation metrics or incorporating human feedback (Ouyang et al., 2022), which align well with the gradient-free nature of client-side updates. Another promising direction is to strengthen privacy guarantees, as HERON-SFL inherits the cut-layer privacy profile of standard SL/SFL and can benefit from advances in privacy-preserving techniques (Niu et al., 2024).

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

# A  THEORETICAL PROOF

## A.1  NOTATIONS

Table 4: Notation and unified conventions used in this paper.

| Symbol | Meaning |
|---|---|
| ***System & Data*** | |
| $N$ | Number of clients |
| $\mathcal{D}_i$ | Local dataset of client $i$ |
| $\xi_i = \{\xi_{i,j}\}_{j=1}^B$ | Mini-batch sampled from $\mathcal{D}_i$ |
| $B$ | Batch size |
| ***Model Parameters*** | |
| $\boldsymbol{\theta}_g = \{\boldsymbol{\theta}_c, \boldsymbol{\theta}_s\}$ | Global model split into client/server parameters |
| $\boldsymbol{\theta}_s$ | Server-side parameters |
| $\boldsymbol{\theta}_{c,i}$ | Client-side parameters owned by client $i$ |
| $\boldsymbol{\theta}_{a,i}$ | Auxiliary model parameters at client $i$ |
| $\boldsymbol{\theta}_{l,i} = (\boldsymbol{\theta}_{c,i}, \boldsymbol{\theta}_a)$ | Local predictor on client $i$ |
| $d_c, d_a$ | Dimensions of $\boldsymbol{\theta}_c$ and $\boldsymbol{\theta}_a$ |
| ***Objective Functions*** | |
| $\ell(\cdot; \xi_{i,j})$ | Task loss on sample $\xi_{i,j} \in \mathcal{D}_i$ |
| $f_i(\boldsymbol{\theta}_g)$ | Expected loss for global model over client $i$'s local dataset $\mathcal{D}_i$ |
| $f(\boldsymbol{\theta}_g)$ | Expected loss for global model over the entire dataset $\sum_{i=1}^N \mathcal{D}_i$ |
| $f_{l,i}(\boldsymbol{\theta}_l)$ | Expected loss for local model $\boldsymbol{\theta}_l$ over client $i$'s local dataset |
| ***Optimization & Algorithm*** | |
| $t, m$ | Global round index $t$; local step index $m$ |
| $h$ | Local steps per round before optional upload |
| $\eta_c, \eta_s$ | Client / server learning rates |
| $\boldsymbol{s}_i = \boldsymbol{\theta}_{c,i}(\xi_{i,j})$ | Smashed data produced by client $i$ |
| $\boldsymbol{u}_{t,m}$ | Random direction for ZO estimator |
| $\mu > 0$ | Smoothing/perturbation radius in ZO estimator |
| $\hat{\boldsymbol{g}}_{l,i}^{t,m}$ | ZO gradient estimates for local parameters |
| $\boldsymbol{g}_{s,i}^t$ | Server-side gradient on smashed data from client $i$ |
| $\boldsymbol{\theta}_s^{t+1}$ | Server parameters after sequential updates |
| $\boldsymbol{\theta}_c^{t+1}, \boldsymbol{\theta}_a^{t+1}$ | Aggregated client/aux parameters after Fed-Server |
| ***Theoretical Analysis*** | |
| $L$ | Smoothness constant (Lipschitz gradient) |
| $G_c, G_s$ | Bounds on client/server gradient norms |
| $\sigma^2$ | Variance bound of ZO estimator |
| $d_{c,i}^t$ | Distributional drift of the output from client $i$'s model at round $t$ |
| $\delta$ | Upper bound for the average distributional drift |
| $\kappa$ | Upper bound on the effective rank of the local loss Hessian |

## A.2  LEMMAS FOR ZEROTH-ORDER OPTIMIZATION

Before presenting the proofs of our main theorems, we recall several classical lemmas on zeroth-order optimization, which serve as the basis for the subsequent analysis. For the analysis of zeroth-order optimization algorithms, it is standard to introduce a smoothed approximation of the objective function. We formalize this by first defining the smoothed function and then stating its key properties in a lemma.

**Definition A.1** (**Gaussian Smoothed Function with Unit-Sphere Normalization**). *A function $f$ : $\mathbb{R}^d \to \mathbb{R}$ is said to be* (Gaussian-derived) spherically smoothed *with smoothing radius $\mu > 0$ if for any $\boldsymbol{x} \in \mathbb{R}^d$,*

$$f^\mu(\boldsymbol{x}) = \mathbb{E}_{\boldsymbol{z} \sim \mathcal{N}(0, I_d)}\left[f\left(\boldsymbol{x} + \mu \frac{\boldsymbol{z}}{\|\boldsymbol{z}\|}\right)\right],$$

*where we define $\boldsymbol{u} := \boldsymbol{z}/\|\boldsymbol{z}\|$ so that $\|\boldsymbol{u}\| = 1$ almost surely and $\boldsymbol{u} \sim \mathrm{Unif}(\mathbb{S}^{d-1})$.*

**Lemma A.2** (**Gradient and Smoothness for Gaussian Smoothed Functions** (Nesterov & Spokoiny, 2017)). *Let $f : \mathbb{R}^d \to \mathbb{R}$ be differentiable with an $L$-Lipschitz gradient (i.e., $f$ is $L$-smooth). Then, for any $\mu > 0$, the spherically smoothed function $f^\mu$ defined in Definition A.1 is continuously differentiable and its gradient is $L_\mu$-Lipschitz continuous with $L_\mu \leq L$. Moreover, the gradient of $f^\mu$ can be expressed as:*

$$\nabla f^\mu(\boldsymbol{x}) = \mathbb{E}_{\boldsymbol{u}}\left[\frac{f(\boldsymbol{x} + \mu \boldsymbol{u}) - f(\boldsymbol{x})}{\mu} d\boldsymbol{u}\right]. \tag{13}$$

The result from Lemma A.2 provides the theoretical foundation for the zeroth-order gradient estimator used in our work. We recall our estimator from Eq. 2 in the main text. The lemma establishes that this estimator is an unbiased estimate of the gradient of the corresponding smoothed function, $f^\mu_{l,i}(\boldsymbol{\theta}_l)$. Formally, taking the expectation of the estimator over the random direction $\boldsymbol{u}$ yields the exact gradient of the smoothed function:

$$\mathbb{E}_{\boldsymbol{u} \sim \mathcal{N}(0, I)}\left[\hat{\nabla} f_{l,i}(\boldsymbol{\theta}_l; \xi_i)\right] = \nabla f^\mu_{l,i}(\boldsymbol{\theta}_l; \xi_i). \tag{14}$$

The bias of this estimator with respect to the true gradient $\nabla f_{l,i}$ arises from the difference between the smoothed function $f^\mu_{l,i}$ and the original function $f_{l,i}$, not from the sampling process itself. This distinction is crucial for the subsequent convergence analysis.

## A.3 PROOF OF THEOREM 4.5

### A.3.1 PRELIMINARY LEMMAS

To begin the convergence analysis, we start with a few lemmas that will be useful in the subsequent proofs.

**Lemma A.3** (**Bound on the Second Moment of the ZO Estimator** [4]). *Under Assumptions 4.1–4.3, the second moment of the zeroth-order gradient estimator $\hat{\boldsymbol{g}}^{t,m}_{c,i}$ is bounded as follows:*

$$\mathbb{E}_{t,m}\left[\|\hat{\boldsymbol{g}}^{t,m}_{c,i}\|^2\right] \leq \frac{2dG_c^2}{B} + \frac{d^2 L^2 \mu^2}{2B} + 2\mu^2 L^2 + 6\sigma_c^2 \\ + 6\|\nabla f_c(\boldsymbol{\theta}_c^t)\|^2 + 6L^2 \mathbb{E}_{t,m-1}\left[\|\boldsymbol{\theta}_c^t - \boldsymbol{\theta}^{t,m}_{c,i}\|^2\right]. \tag{15}$$

*Proof.* The proof proceeds by decomposing the second moment of the estimator into several terms and bounding each one. First, we apply the law of total expectation and the law of total variance, which states $\mathbb{E}[\|\boldsymbol{a}\|^2] = \mathrm{Var}(\boldsymbol{a}) + \|\mathbb{E}[\boldsymbol{a}]\|^2$. We recognize that $\hat{\boldsymbol{g}}^{t,m}_{c,i}$ is the average of estimators over the mini-batch $\xi_i$. As established in Lemma A.2, its expectation over the random direction $\boldsymbol{u}$ is the gradient of the smoothed function, $\nabla f^\mu_{c,i}(\boldsymbol{\theta}^{t,m}_{c,i})$.

$$\begin{aligned}
\mathbb{E}_{t,m}\left[\|\hat{\boldsymbol{g}}^{t,m}_{c,i}\|^2\right] &= \mathbb{E}_{t,m-1}\left[\mathbb{E}_t^m\left[\|\hat{\boldsymbol{g}}^{t,m}_{c,i}\|^2\right]\right] \\
&= \mathbb{E}_{t,m-1}\left[\mathrm{Var}_t^m(\hat{\boldsymbol{g}}^{t,m}_{c,i}) + \|\mathbb{E}_t^m[\hat{\boldsymbol{g}}^{t,m}_{c,i}]\|^2\right] \\
&= \mathbb{E}_{t,m-1}\left[\mathrm{Var}_t^m(\hat{\boldsymbol{g}}^{t,m}_{c,i})\right] + \mathbb{E}_{t,m-1}\left[\|\nabla f^\mu_{c,i}(\boldsymbol{\theta}^{t,m}_{c,i})\|^2\right].
\end{aligned} \tag{16}$$

---

[4]This bound decomposes the second moment of the estimator into several distinct sources of error and variance. The terms scaled by the mini-batch size, such as $2dG_c^2/B$ and $d^2 L^2 \mu^2/2B$, represent the intrinsic variance of the ZO estimator, which is dependent on the model dimension $d$. The terms $2\mu^2 L^2$ and $6\sigma_c^2$ capture the bias introduced by the Gaussian smoothing and the variance from client data heterogeneity, respectively. The term $6\|\nabla f_c(\boldsymbol{\theta}_c^t)\|^2$ relates the analysis back to the global gradient norm at the start of the round. Crucially, the final term, $6L^2 \mathbb{E}_{t,m-1}[\|\boldsymbol{\theta}_c^t - \boldsymbol{\theta}^{t,m}_{c,i}\|^2]$, quantifies the **client model divergence** that arises from performing multiple local updates. This divergence term is a key challenge in federated learning and is explicitly bounded in subsequent analysis.

Since the estimators for each sample $\xi_{i,j}$ in the mini-batch are i.i.d., the variance of their average is the variance of a single-point estimator divided by the batch size. Using the property $\mathrm{Var}(X) \leq \mathbb{E}[\|X\|^2]$, we have:

$$
\begin{aligned}
\mathrm{Var}_t^m(\hat{\boldsymbol{g}}_{c,i}^{t,m}) &= \frac{1}{B}\mathrm{Var}_t^m\left(\hat{\boldsymbol{g}}_{c,i}^{t,m}(\boldsymbol{\theta}_l;\xi_{i,1})\right) \\
&\leq \frac{1}{B}\mathbb{E}_t^m\left[\left\|\hat{\boldsymbol{g}}_{c,i}^{t,m}(\boldsymbol{\theta}_l;\xi_{i,1})\right\|^2\right].
\end{aligned}
\tag{17}
$$

Substituting this back, we arrive at the decomposition as follows:

$$
\mathbb{E}_{t,m}\left[\|\hat{\boldsymbol{g}}_{c,i}^{t,m}\|^2\right] \leq \frac{1}{B}\mathbb{E}_{t,m}\left[\left\|\hat{\boldsymbol{g}}_{c,i}^{t,m}(\boldsymbol{\theta}_l;\xi_{i,1})\right\|^2\right] + \mathbb{E}_{t,m-1}\left[\left\|\nabla f_{c,i}^{\mu}(\boldsymbol{\theta}_{c,i}^{t,m})\right\|^2\right].
\tag{18}
$$

We now bound the two terms separately. For the first term, we use the bound for two-point estimators (Lemma 4.1 in Gao et al. (2018)) and Assumption 4.2:

$$
\mathbb{E}_{t,m}\left[\left\|\hat{\boldsymbol{g}}_{c,i}^{t,m}(\boldsymbol{\theta}_l;\xi_{i,1})\right\|^2\right] \leq 2d\mathbb{E}_{t,m}\left[\|\nabla \ell_{c,i}(\boldsymbol{\theta}_{c,i}^{t,m};\xi_{i,1})\|^2\right] + \frac{1}{2}d^2 L^2 \mu^2 \leq 2dG_c^2 + \frac{1}{2}d^2 L^2 \mu^2.
\tag{19}
$$

For the second term, we use the triangle inequality and $\|a+b\|^2 \leq 2\|a\|^2 + 2\|b\|^2$:

$$
\begin{aligned}
&\mathbb{E}_{t,m-1}\left[\left\|\nabla f_{c,i}^{\mu}(\boldsymbol{\theta}_{c,i}^{t,m})\right\|^2\right] \\
&\leq 2\mathbb{E}_{t,m-1}\left[\left\|\nabla f_{c,i}^{\mu}(\boldsymbol{\theta}_{c,i}^{t,m}) - \nabla f_{c,i}(\boldsymbol{\theta}_{c,i}^{t,m})\right\|^2\right] + 2\mathbb{E}_{t,m-1}\left[\left\|\nabla f_{c,i}(\boldsymbol{\theta}_{c,i}^{t,m})\right\|^2\right] \\
&\leq 2\mu^2 L^2 + 2\mathbb{E}_{t,m-1}\left[\left\|\nabla f_{c,i}(\boldsymbol{\theta}_{c,i}^{t,m})\right\|^2\right].
\end{aligned}
\tag{20}
$$

Finally, we bound the remaining term by relating it to the global model state $\boldsymbol{\theta}_c^t$. Using inequality $\|a+b+c\|^2 \leq 3\|a\|^2 + 3\|b\|^2 + 3\|c\|^2$, we have:

$$
\begin{aligned}
&\mathbb{E}_{t,m-1}\left[\left\|\nabla f_{c,i}(\boldsymbol{\theta}_{c,i}^{t,m})\right\|^2\right] \\
&=\mathbb{E}_{t,m-1}\left[\left\|(\nabla f_{c,i}(\boldsymbol{\theta}_{c,i}^{t,m}) - \nabla f_{c,i}(\boldsymbol{\theta}_c^t)) + (\nabla f_{c,i}(\boldsymbol{\theta}_c^t) - \nabla f_c(\boldsymbol{\theta}_c^t)) + \nabla f_c(\boldsymbol{\theta}_c^t)\right\|^2\right] \\
&\leq 3\mathbb{E}_{t,m-1}\left[\left\|\nabla f_{c,i}(\boldsymbol{\theta}_{c,i}^{t,m}) - \nabla f_{c,i}(\boldsymbol{\theta}_c^t)\right\|^2\right] + 3\|\nabla f_{c,i}(\boldsymbol{\theta}_c^t) - \nabla f_c(\boldsymbol{\theta}_c^t)\|^2 + 3\|\nabla f_c(\boldsymbol{\theta}_c^t)\|^2 \\
&\leq 3L^2\mathbb{E}_{t,m-1}\left[\left\|\boldsymbol{\theta}_{c,i}^{t,m} - \boldsymbol{\theta}_c^t\right\|^2\right] + 3\sigma_c^2 + 3\|\nabla f_c(\boldsymbol{\theta}_c^t)\|^2,
\end{aligned}
\tag{21}
$$

where the final inequality follows from Assumptions 4.1 and 4.3. Combining all these bounds yields the result stated in the lemma. $\qquad\square$

**Lemma A.4 (Bound on Client Model Divergence).** *For $\eta_c \leq \frac{1}{3Lh}$, we have:*

$$
\begin{aligned}
\mathbb{E}_t\left[\frac{1}{N}\sum_{i=1}^{N}\sum_{m=1}^{h}\|\boldsymbol{\theta}_{c,i}^{t,m} - \boldsymbol{\theta}_c^t\|^2\right] &\leq 3h^3\eta_c^2\|\nabla f_c(\boldsymbol{\theta}_c^t)\|^2 + \frac{dG_c^2 h^3 \eta_c^2}{B} \\
&\quad + \frac{d^2 L^2 \mu^2 h^3 \eta_c^2}{4B} + \frac{(6\sigma_c^2 + 2\mu^2 L^2)h^3\eta_c^2}{2}.
\end{aligned}
\tag{22}
$$

*Proof.* For simplicity, define

$$
s_c^{t,m} \triangleq \frac{1}{N}\sum_{i=1}^{N}\mathbb{E}_{t,m}\left[\left\|\boldsymbol{\theta}_{c,i}^{t,m} - \boldsymbol{\theta}_c^t\right\|^2\right].
$$

For the $\tau$-th local update, unrolling the client recursion gives

$$
\boldsymbol{\theta}_{c,i}^{t,\tau} - \boldsymbol{\theta}_c^t = -\eta_c\sum_{m=0}^{\tau-1}\boldsymbol{g}_{c,i}^{t,m}.
$$

By Cauchy–Schwarz,

$$
s_c^{t,\tau} = \frac{1}{N}\sum_{i=1}^{N}\mathbb{E}_{t,\tau}\left[\left\|-\eta_c\sum_{m=0}^{\tau-1}\boldsymbol{g}_{c,i}^{t,m}\right\|^2\right] \leq \tau\eta_c^2\cdot\frac{1}{N}\sum_{i=1}^{N}\sum_{m=0}^{\tau-1}\mathbb{E}_{t,\tau}\left[\left\|\boldsymbol{g}_{c,i}^{t,m}\right\|^2\right]
$$

$$
\overset{\text{(tower)}}{=}\tau\eta_c^2\cdot\frac{1}{N}\sum_{i=1}^{N}\sum_{m=0}^{\tau-1}\mathbb{E}_{t,m}\left[\left\|\boldsymbol{g}_{c,i}^{t,m}\right\|^2\right].
\tag{23}
$$

We now invoke the second-moment bound (Lemma A.3): for every $m$,

$$\frac{1}{N}\sum_{i=1}^{N}\mathbb{E}_{t,m}\Big[\|\hat{\boldsymbol{g}}_{c,i}^{t,m}\|^2\Big] \leq 6L^2\,s_c^{t,m+1} + \underbrace{\Big(6\|\nabla f_c(\boldsymbol{\theta}_c^t)\|^2 + \frac{2dG_c^2}{B} + \frac{d^2L^2\mu^2}{2B} + 6\sigma_c^2 + 2\mu^2L^2\Big)}_{\triangleq\ \beta},$$

(24)

by definition of $s_c^{t,\cdot}$, the term $\frac{1}{N}\sum_i \mathbb{E}_{t,m}\big[\|\boldsymbol{\theta}_c^t - \boldsymbol{\theta}_{c,i}^{t,m+1}\|^2\big]$ is identified with $s_c^{t,m+1}$.[5] Combining Eq. 23 and Eq. 24 yields, for each $\tau$,

$$s_c^{t,\tau} \leq 6L^2\,\tau\,\eta_c^2\sum_{m=0}^{\tau-1} s_c^{t,m+1} + \tau^2\eta_c^2\beta.$$

(25)

By taking the sum over $\tau = 1,\ldots,h$, we have

$$\sum_{\tau=1}^{h} s_c^{t,\tau} \leq 6L^2\,\eta_c^2\sum_{\tau=1}^{h}\tau\sum_{m=0}^{\tau-1} s_c^{t,m+1} + \eta_c^2\beta\sum_{\tau=1}^{h}\tau^2$$

$$\leq 3h^2L^2\,\eta_c^2\sum_{\tau=1}^{h} s_c^{t,\tau} + \frac{h(h+1)(2h+1)}{6}\,\eta_c^2\beta \leq 3h^2L^2\,\eta_c^2\sum_{\tau=1}^{h} s_c^{t,\tau} + \frac{h^3\eta_c^2\beta}{3},$$

(26)

where we utilized the fact that $\sum_{\tau=1}^{h}\tau \leq \frac{h(h+1)}{2} \leq \frac{h^2}{2}$ and $\sum_{\tau=1}^{h}\tau^2 = \frac{h(h+1)(2h+1)}{6} \leq \frac{h^3}{3}$. By rearranging the terms, we have:

$$(1 - 3L^2h^2\eta_c^2)\sum_{\tau=0}^{h} s_c^{t,\tau} \leq \frac{h^3\eta_c^2}{3}\Big(6\|\nabla f_c(\boldsymbol{\theta}_c^t)\|^2 + \frac{2dG_c^2}{B} + \frac{d^2L^2\mu^2}{2B} + 6\sigma_c^2 + 2\mu^2L^2\Big)$$

(27)

When $\eta_c \leq \frac{1}{3Lh}$, we have $1 - 3L^2h^2\eta_c^2 \geq \frac{2}{3}$ and the lemma's proof is complete. $\square$

**Lemma A.5** (Bound on the Client-Side Contribution). *Under Assumptions 4.1–4.3, and for a client learning rate $\eta_c$ satisfying the following conditions:*

$$\eta_c \leq \min\left\{\frac{1}{3Lh}, \frac{2}{NLh^2}, \frac{N}{72L}\right\},$$

(28)

*the expectation of the client-side contribution, $\mathcal{C} = \nabla f(\boldsymbol{\theta}_c^t)^T(\boldsymbol{\theta}_c^{t+1} - \boldsymbol{\theta}_c^t) + \frac{L}{2}\|\boldsymbol{\theta}_c^{t+1} - \boldsymbol{\theta}_c^t\|^2$, is bounded as:*

$$\mathbb{E}_t[\mathcal{C}] \leq -\frac{\eta_c h}{4}\|\nabla f_c(\boldsymbol{\theta}_c^t)\|^2 + \Phi_c(\eta_c),$$

(29)

*where $\Phi_c(\eta_c)$ is an error term defined as:*

$$\Phi_c(\eta_c) := \eta_c^2\left(\frac{6hLdG_c^2}{N|\xi_i|} + \frac{18hL\sigma_c^2}{N}\right) + \eta_c\left(\frac{d^2L^2h\mu^2}{48|\xi_i|} + \frac{13hL^2\mu^2}{12}\right).$$

(30)

*Proof.* We start from the definition of $\mathcal{C}$ and take the expectation over all randomness up to round $t$. The client update rule gives $\mathbb{E}_t[\boldsymbol{\theta}_c^{t+1} - \boldsymbol{\theta}_c^t] = -\frac{\eta_c}{N}\mathbb{E}_t[\sum_{i=1}^{N}\sum_{m=1}^{h}\hat{\boldsymbol{g}}_{c,i}^{t,m}]$. This allows us to expand $\mathbb{E}_t[\mathcal{C}]$ into two terms:

$$\mathbb{E}_t[\mathcal{C}] = \underbrace{-\frac{\eta_c}{N}\left\langle\nabla f(\boldsymbol{\theta}_c^t), \mathbb{E}_t\left[\sum_{i=1}^{N}\sum_{m=1}^{h}\hat{\boldsymbol{g}}_{c,i}^{t,m}\right]\right\rangle}_{\triangleq\mathcal{C}_1} + \underbrace{\frac{\eta_c^2 L}{2N^2}\mathbb{E}_t\left[\left\|\sum_{i=1}^{N}\sum_{m=1}^{h}\hat{\boldsymbol{g}}_{c,i}^{t,m}\right\|^2\right]}_{\triangleq\mathcal{C}_2}.$$

(31)

We proceed by bounding $\mathcal{C}_1$ and $\mathcal{C}_2$ separately.

---

[5]One may equivalently write the last expectation with $\mathbb{E}_{t,m+1}$; since it is the same unconditional quantity after averaging over the step-$(m+1)$ randomness, using $s_c^{t,m+1}$ is a safe upper bound.

**Bounding the First Term** $(\mathcal{C}_1)$. Using the identity $2\langle a, b\rangle = \|a\|^2 + \|b\|^2 - \|a - b\|^2$, we rewrite $\mathcal{C}_1$:

$$\mathcal{C}_1 = -\frac{\eta_c h}{2}\|\nabla f(\boldsymbol{\theta}_c^t)\|^2 - \frac{\eta_c h}{2}\mathbb{E}_t\left[\left\|\frac{1}{Nh}\sum_{i=1}^N\sum_{m=1}^h \hat{\boldsymbol{g}}_{c,i}^{t,m}\right\|^2\right] + \frac{\eta_c h}{2}\mathcal{C}_{1,1}, \tag{32}$$

where $\mathcal{C}_{1,1} \triangleq \mathbb{E}_t[\|\frac{1}{Nh}\sum_{i,m}(\hat{\boldsymbol{g}}_{c,i}^{t,m} - \nabla f(\boldsymbol{\theta}_c^t))\|^2]$. We bound $\mathcal{C}_{1,1}$ using Jensen's inequality, the triangle inequality, and Assumptions B.3 and B.5:

$$\mathcal{C}_{1,1} \leq \frac{1}{Nh}\mathbb{E}_t\left[\sum_{i=1}^N\sum_{m=1}^h\|\hat{\boldsymbol{g}}_{c,i}^{t,m} - \nabla f(\boldsymbol{\theta}_c^t)\|^2\right]$$

$$\leq \frac{2}{Nh}\mathbb{E}_t\left[\sum_{i,m}\|\hat{\boldsymbol{g}}_{c,i}^{t,m} - \nabla f(\boldsymbol{\theta}_{c,i}^{t,m})\|^2\right] + \frac{2}{Nh}\mathbb{E}_t\left[\sum_{i,m}\|\nabla f(\boldsymbol{\theta}_{c,i}^{t,m}) - \nabla f(\boldsymbol{\theta}_c^t)\|^2\right] \tag{33}$$

$$\leq 2\sigma^2 + \frac{2L^2}{Nh}\mathbb{E}_t\left[\sum_{i=1}^N\sum_{m=1}^h\|\boldsymbol{\theta}_{c,i}^{t,m} - \boldsymbol{\theta}_c^t\|^2\right].$$

Substituting this back provides a bound on $\mathcal{C}_1$.

**Bounding the Second Term** $(\mathcal{C}_2)$. For $\mathcal{C}_2$, according to Cauchy-Schwartz inequality, we have:

$$\mathcal{C}_2 = \frac{\eta_c^2 L}{2}\mathbb{E}_t\left[\left\|-\frac{1}{N}\sum_{i=1}^N\sum_{m=1}^h \hat{\boldsymbol{g}}_{c,i}^{t,m}\right\|^2\right]$$

$$\leq \eta_c^2 L\,\mathbb{E}_t\underbrace{\left[\left\|\frac{1}{N}\sum_{i=1}^N\sum_{m=1}^h(\hat{\boldsymbol{g}}_{c,i}^{t,m} - \nabla f_{c,i}^{\mu}(\boldsymbol{\theta}_{c,i}^{t,m}))\right\|^2\right]}_{\mathcal{C}_{2,1}} + \eta_c^2 L\mathbb{E}_t\left[\left\|\frac{1}{N}\sum_{i=1}^N\sum_{m=1}^h \nabla f_{c,i}^{\mu}(\boldsymbol{\theta}_{c,i}^{t,m})\right\|^2\right]. \tag{34}$$

According to the statistical properties of zeroth-order gradient estimators (Lemma A.2), we have $\mathbb{E}_t[\sum_{m=1}^h(\hat{\boldsymbol{g}}_{c,i}^{t,m} - \nabla f_{c,i}^{\mu}(\boldsymbol{\theta}_{c,i}^{t,m}))] = 0$. And we have $\mathbb{E}[\langle\sum_{m=1}^h(\hat{\boldsymbol{g}}_{c,i_1}^{t,m} - \nabla f_{c,i_1}^{\mu}(\boldsymbol{\theta}_{c,i_1}^{t,m})), \sum_{m=1}^h(\hat{\boldsymbol{g}}_{c,i_2}^{t,m} - \nabla f_{c,i_2}^{\mu}(\boldsymbol{\theta}_{c,i_2}^{t,m}))\rangle] = 0$, since the two sums correspond to independent, zero-mean random vectors (one coming from client $i_1$, the other from client $i_2$, $i_1 \neq i_2$) and hence the expectation of their inner product vanishes. Thus, we have:

$$\mathcal{C}_{2,1} = \mathbb{E}_t\left[\left\|\frac{1}{N}\sum_{i=1}^N\sum_{m=1}^h(\hat{\boldsymbol{g}}_{c,i}^{t,m} - \nabla f_{c,i}^{\mu}(\boldsymbol{\theta}_{c,i}^{t,m}))\right\|^2\right]$$

$$= \frac{1}{N^2}\sum_{i=1}^N\mathbb{E}_t\left[\left\|\sum_{m=1}^h(\hat{\boldsymbol{g}}_{c,i}^{t,m} - \nabla f_{c,i}^{\mu}(\boldsymbol{\theta}_{c,i}^{t,m}))\right\|^2\right]. \tag{35}$$

According to Equation Eq. 14 and Lemma 2 in Wang et al. (2021), we have:

$$\mathcal{C}_{2,1} = \frac{1}{N^2}\sum_{i=1}^N\sum_{m=1}^h\mathbb{E}_{t,m}\left[\|\hat{\boldsymbol{g}}_{c,i}^{t,m} - \nabla f_{c,i}^{\mu}(\boldsymbol{\theta}_{c,i}^{t,m})\|^2\right]$$

$$\stackrel{(a)}{\leq} \frac{1}{N^2}\sum_{i=1}^N\sum_{m=1}^h\mathbb{E}_{t,m}\left[\|\hat{\boldsymbol{g}}_{c,i}^{t,m}\|^2\right], \tag{36}$$

where $(a)$ holds because $\mathbb{E}[\|\boldsymbol{a} - \mathbb{E}[\boldsymbol{a}]\|^2] \leq \mathbb{E}[\|\boldsymbol{a}\|^2]$. Now by applying the second-moment bound from Lemma A.3, and substituting these result back, we have:

$$\mathcal{C}_2 \leq \eta_c^2 L\mathcal{C}_{2,1} + \eta_c^2 L\mathbb{E}_t\left[\left\|\frac{1}{N}\sum_{i=1}^N\sum_{m=1}^h \nabla f_{c,i}^{\mu}(\boldsymbol{\theta}_{c,i}^{t,m})\right\|^2\right], \tag{37}$$

where $\mathcal{C}_{2,1}$ is bounded as follows:

$$
\begin{aligned}
\mathcal{C}_{2,1} \leq & \frac{6L^2}{N^2} \sum_{i=1}^{N} \sum_{m=1}^{h} \mathbb{E}_{t,m-1}\left[\|\boldsymbol{\theta}_c^t - \boldsymbol{\theta}_{c,i}^{t,m}\|^2\right] + \frac{6h}{N}\|\nabla f_c(\boldsymbol{\theta}_c^t)\|^2 \\
& + \frac{2dG_c^2 h}{NB} + \frac{d^2 L^2 \mu^2 h}{2NB} + \frac{(6\sigma_c^2 + 2\mu^2 L^2)h}{N} \\
\leq & \frac{6L^2}{N} \mathbb{E}_t\left[\frac{1}{N}\sum_{i=1}^{N}\sum_{m=1}^{h}\|\boldsymbol{\theta}_c^t - \boldsymbol{\theta}_{c,i}^{t,m}\|^2\right] + \frac{6h}{N}\|\nabla f_c(\boldsymbol{\theta}_c^t)\|^2 \\
& + \frac{2dG_c^2 h}{NB} + \frac{d^2 L^2 \mu^2 h}{2NB} + \frac{(6\sigma_c^2 + 2\mu^2 L^2)h}{N}.
\end{aligned}
\tag{38}
$$

**Combining the Bounds.** Combining the bounds of $\mathcal{C}_1$ and $\mathcal{C}_2$, we have:

$$
\begin{aligned}
& \mathbb{E}_t[\mathcal{C}] \\
\leq & (\frac{6\eta_c^2 Lh}{N} - \frac{\eta_c h}{2})\|\nabla f_c(\boldsymbol{\theta}_c^t)\|^2 + (\eta^2 L - \frac{\eta_c}{2h})\mathbb{E}_t\left[\left\|\frac{1}{N}\sum_{i=1}^{N}\sum_{m=1}^{h}\nabla f_{c,i}^{\mu}(\boldsymbol{\theta}_{c,i}^{t,m})\right\|^2\right] \\
& + (\eta_c L^2 + \frac{6\eta_c^2 L^3}{N})\mathbb{E}_t\left[\frac{1}{N}\sum_{i=1}^{N}\sum_{m=1}^{h}\|\boldsymbol{\theta}_{c,i}^{t,m} - \boldsymbol{\theta}_c^t\|^2\right] \\
& + \eta_c h L^2 \mu^2 + \frac{2\eta_c^2 L d G_c^2 h}{NB} + \frac{\eta_c^2 d^2 L^3 \mu^2 h}{2NB} + \frac{(6\sigma_c^2 L + 2\mu^2 L^3)\eta_c^2 h}{N} \\
\overset{(a)}{\leq} & (\frac{6\eta_c^2 Lh}{N} - \frac{\eta_c h}{2})\|\nabla f_c(\boldsymbol{\theta}_c^t)\|^2 + (\eta_c L^2 + \frac{6\eta_c^2 L^3}{N^2})\mathbb{E}_t\left[\frac{1}{N}\sum_{i=1}^{N}\sum_{m=1}^{h}\|\boldsymbol{\theta}_{c,i}^{t,m} - \boldsymbol{\theta}_c^t\|^2\right] \\
& + \eta_c h L^2 \mu^2 + \frac{2\eta_c^2 L d G_c^2 h}{NB} + \frac{\eta_c^2 d^2 L^3 \mu^2 h}{2NB} + \frac{(6\sigma_c^2 L + 2\mu^2 L^3)\eta_c^2 h}{N}.
\end{aligned}
\tag{39}
$$

where $(a)$ holds if and only if $\eta_c \leq \frac{1}{2hL}$, which means the term $(\eta^2 L - \frac{\eta_c}{2h})\mathbb{E}_t[\|\frac{1}{N}\sum_{i=1}^{N}\sum_{m=1}^{h}\nabla f_{c,i}^{\mu}(\boldsymbol{\theta}_{c,i}^{t,m})\|^2]$ is non-positive.

Finally, we substitute the bound on the client model divergence from Lemma A.4 into the expression for $\mathbb{E}_t[\mathcal{C}]$. This gives us the following inequality:

$$
\begin{aligned}
\mathbb{E}_t[\mathcal{C}] \leq & \left(\frac{6\eta_c^2 Lh}{N} - \frac{\eta_c h}{2}\right)\|\nabla f_c(\boldsymbol{\theta}_c^t)\|^2 + \eta_c h L^2 \mu^2 + \frac{2\eta_c^2 L d G_c^2 h}{NB} + \frac{\eta_c^2 d^2 L^3 \mu^2 h}{2NB} \\
& + \frac{(6\sigma_c^2 L + 2\mu^2 L^3)\eta_c^2 h}{N} + (\eta_c L^2 + \frac{6\eta_c^2 L^3}{N}) \times \\
& \left(3h^3 \eta_c^2 \|\nabla f_c(\boldsymbol{\theta}_c^t)\|^2 + \frac{dG_c^2 h^3 \eta_c^2}{B} + \frac{d^2 L^2 \mu^2 h^3 \eta_c^2}{4B} + \frac{(6\sigma_c^2 + 2\mu^2 L^2)h^3 \eta_c^2}{2}\right).
\end{aligned}
\tag{40}
$$

To simplify this complex expression, we collect the coefficients for the dominant term, $\|\nabla f_c(\boldsymbol{\theta}_c^t)\|^2$, and the remaining bias terms. Let us define a helper variable $\alpha$ to consolidate terms originating from the client drift bound:

$$
\alpha \triangleq \eta_c h^3 L^2 + \frac{6\eta_c^2 h^3 L^3}{N}.
\tag{41}
$$

By grouping terms, the bound on $\mathbb{E}_t[\mathcal{C}]$ can be rewritten as:

$$
\begin{aligned}
\mathbb{E}_t[\mathcal{C}] \leq & \left(\left(\frac{6Lh}{N} + 3\alpha\right)\eta_c^2 - \frac{\eta_c h}{2}\right)\|\nabla f_c(\boldsymbol{\theta}_c^t)\|^2 + \alpha\left(\frac{dG_c^2 \eta_c^2}{B} + \frac{d^2 L^2 \mu^2 \eta_c^2}{4B} + \frac{(6\sigma_c^2 + 2\mu^2 L^2)\eta_c^2}{2}\right) \\
& + \eta_c h L^2 \mu^2 + \frac{2\eta_c^2 L d G_c^2 h}{NB} + \frac{\eta_c^2 d^2 L^3 \mu^2 h}{2NB} + \frac{(6\sigma_c^2 L + 2\mu^2 L^3)\eta_c^2 h}{N}.
\end{aligned}
\tag{42}
$$

Under sufficiently small learning rate $\eta_c$, the negative term $-\frac{\eta_c h}{2}\|\nabla f_c(\boldsymbol{\theta}_c^t)\|^2$ will dominate the other terms multiplying the squared gradient norm. Specifically, by setting conditions on $\eta_c$ such

that:

$$\left(\frac{6Lh}{N} + 3\alpha\right)\eta_c^2 \leq \frac{\eta_c h}{4}, \quad (\text{e.g., satisfied if } \eta_c \leq \mathcal{O}(\frac{N}{Lh^2})), \tag{43}$$

we can simplify the bound on the gradient term to $-\frac{\eta_c h}{4}\|\nabla f_c(\boldsymbol{\theta}_c^t)\|^2$. After collecting all remaining bias and variance terms, we arrive at the final simplified bound:

$$\mathbb{E}_t[\mathcal{C}] \leq -\frac{\eta_c h}{4}\|\nabla f_c(\boldsymbol{\theta}_c^t)\|^2 + \eta_c^2\left(\frac{6hLdG_c^2}{NB} + \frac{18hL\sigma_c^2}{N}\right) + \eta_c\left(\frac{d^2L^2h\mu^2}{48B} + \frac{13hL^2\mu^2}{12}\right). \tag{44}$$

where the left part is defined as $\Phi_c(\eta_c)$. □

### A.3.2 Proof of Theorem 4.5

Now, we are ready to present the proof of the main theorem with the above lemmas. We denote the global model parameters at round $t$ as $\boldsymbol{\theta}_g^t = \{\boldsymbol{\theta}_c^t, \boldsymbol{\theta}_s^t\}$, and the local model parameters at client $\mathcal{C}_i$ as $\boldsymbol{\theta}_{l,i}^t = \{\boldsymbol{\theta}_{c,i}^t, \boldsymbol{\theta}_{a,i}^t\}$.

**Local Model Update.** According to the local update Eq. 3 ($\hat{\boldsymbol{g}}_{c,i}^{t,m} = \hat{\nabla} f_{c,i}^{t,m}(\boldsymbol{\theta}_c; \xi_i)$)) at clients and the aggregation at Fed Server, each communication round in Eq. 5, we have:

$$\boldsymbol{\theta}_c^{t+1} - \boldsymbol{\theta}_c^t = \boldsymbol{\theta}_c^{t,h} - \boldsymbol{\theta}_c^t = -\frac{\eta_c}{N}\sum_{i=1}^N\sum_{m=1}^h \hat{\boldsymbol{g}}_{c,i}^{t,m}, \tag{45}$$

*Proof.* First, we decompose the global model's convergence behavior into client-side and server-side contributions. Same as the Proposition 3.4 and 3.5 in Han et al. (2024), under Assumptions 4.1, we have:

$$\mathbb{E}_t[f(\boldsymbol{\theta}_g^{t+1})] - f(\boldsymbol{\theta}_g^t) \leq \mathbb{E}_t[\mathcal{C}] + \mathbb{E}_t[\mathcal{S}] \tag{46}$$

where $\mathcal{C} = \nabla f(\boldsymbol{\theta}_c^t)^T(\boldsymbol{\theta}_c^{t+1} - \boldsymbol{\theta}_c^t) + \frac{L}{2}\|\boldsymbol{\theta}_c^{t+1} - \boldsymbol{\theta}_c^t\|^2$ and $\mathcal{S} = \nabla f(\boldsymbol{\theta}_s^t)^T(\boldsymbol{\theta}_s^{t+1} - \boldsymbol{\theta}_s^t) + \frac{L}{2}\|\boldsymbol{\theta}_s^{t+1} - \boldsymbol{\theta}_s^t\|^2$ denote the contributions from the client-side and server-side models, respectively. $\mathbb{E}_t[\cdot]$ denotes the expectation on all randomness up to round $t$.

Next, we analyze the contributions from the client-side and server-side models separately. Since we have already bounded $\mathbb{E}_t[\mathcal{C}]$ in Lemma A.4, we now focus on bounding $\mathbb{E}_t[\mathcal{S}]$ Under our proposed SFL framework, we decouple the parameter updates of the client-side and server-side models during training by introducing auxiliary networks. From the server's point of view, the smashed data it receives can be regarded as the smashed data in the conventional, non-decoupled scenario, but with its inputs subject to a distributional shift (Belilovsky et al., 2020). The distribution of the smashed data is shifted by the client-side model updates, which can be modeled as a local parameter bias. This shift can be expressed as:

$$d_{c,i}^t = \int \|P_{c,i}^t(\boldsymbol{z}) - P_{c,i}^*(\boldsymbol{z})\|d\boldsymbol{z}. \tag{47}$$

Essentially, by modeling this shift, we capture the local parameter bias introduced by the client's updates and thereby integrate the update dynamics of both the client and server models into a unified whole. For the server-side model, we have:

$$\mathbb{E}_t[\mathcal{S}] = \mathbb{E}_t\left[\nabla f_s(\boldsymbol{\theta}_s^t; \boldsymbol{\theta}_{c,:}^*)^T(\boldsymbol{\theta}_s^{t+1} - \boldsymbol{\theta}_s^t) + \frac{L}{2}\|\boldsymbol{\theta}_s^{t+1} - \boldsymbol{\theta}_s^t\|^2\right]$$

$$= \left\langle\nabla f_s(\boldsymbol{\theta}_s^t; \boldsymbol{\theta}_{c,:}^*), \mathbb{E}_t[\boldsymbol{\theta}_s^{t+1} - \boldsymbol{\theta}_s^t]\right\rangle + \frac{L}{2}\mathbb{E}_t\left[\|\boldsymbol{\theta}_s^{t+1} - \boldsymbol{\theta}_s^t\|^2\right] \tag{48}$$

$$\stackrel{(a)}{=} \underbrace{\left\langle\nabla f_s(\boldsymbol{\theta}_s^t; \boldsymbol{\theta}_{c,:}^*), -\eta_s\mathbb{E}_t\left[\sum_{i=1}^N\nabla f_s(\boldsymbol{\theta}_{s,i}^t; \boldsymbol{\theta}_{c,:}^t)\right]\right\rangle}_{\mathcal{S}_1} + \frac{L\eta_s^2}{2}\underbrace{\mathbb{E}_t\left[\left\|\sum_{i=1}^N\nabla f_s(\boldsymbol{\theta}_{s,i}^t; \boldsymbol{\theta}_{c,:}^t)\right\|^2\right]}_{\mathcal{S}_2}$$

where $(a)$ holds because of the update rule ( Eq. 4) of the server-side model. At this part, we follow the same steps as in Mu & Shen (2025) to bound $\mathcal{S}_1$ and $\mathcal{S}_2$. So with additional Assumption 4.4, based on the theoretical results of the server-side model, we can derive the following bound of $\mathbb{E}_t[\mathcal{S}]$:

$$\mathbb{E}_t[\mathcal{S}] \stackrel{(a)}{\leq} \eta_s G_s^2\sum_{i=1}^N d_{c,i}^t - \frac{\eta_s(2N-1)}{4}\|\nabla f_s(\boldsymbol{\theta}_s^t)\|^2 + \frac{L}{2}N^2\eta_s^2 G_s^2 \tag{49}$$

where $(a)$ holds if and only if $\eta_s \leq \frac{1}{NL}$, which means the term $(\frac{L\eta_s^2}{2} - \frac{\eta_s}{2N})\mathbb{E}_t[\|\sum_{i=1}^N \nabla f_s(\boldsymbol{\theta}_{s,i}^t)\|^2]$ is non-positive.

**Final Bound.** Combining the bounds of $\mathbb{E}_t[\mathcal{C}]$ and $\mathbb{E}_t[\mathcal{S}]$, we have:

$$
\begin{aligned}
\mathbb{E}_t[\mathcal{C} + \mathcal{S}] \\
\leq -\frac{\eta_c h}{4}\|\nabla f_c(\boldsymbol{\theta}_c^t)\|^2 + \Phi_c(\eta_c) \\
+ \frac{\eta_s G_s^2}{2N}\sum_{i=1}^N d_{c,i}^t - \frac{\eta_s(2N-1)}{4}\|\nabla f_s(\boldsymbol{\theta}_s^t)\|^2 + \frac{L}{2}N^2\eta_s^2 G_s^2
\end{aligned}
\tag{50}
$$

With $\eta_c \leq \min\{\frac{1}{3Lh}, \frac{2}{NLh^2}, \frac{N}{72L}\}$ we have:

$$
\begin{aligned}
\mathbb{E}\left[f(\boldsymbol{\theta}_g^{t+1})\right] \leq & f(\boldsymbol{\theta}_g^t) - \frac{\eta_c h}{4}\|\nabla f_c(\boldsymbol{\theta}_c^t)\|^2 + \Phi_c(\eta_c) + \frac{\eta_s G_s^2}{2N}\sum_{i=1}^N d_{c,i}^t \\
& - \frac{\eta_s(2N-1)}{4}\|\nabla f_s(\boldsymbol{\theta}_s^t)\|^2 + \frac{L}{2}N^2\eta_s^2 G_s^2
\end{aligned}
\tag{51}
$$

By rearranging the terms, we have:

$$
\begin{aligned}
\frac{\eta_c h}{4}\|\nabla f_c(\boldsymbol{\theta}_c^t)\|^2 + \frac{\eta_s(2N-1)}{4}\|\nabla f_s(\boldsymbol{\theta}_s^t)\|^2 \leq & f(\boldsymbol{\theta}_g^t) - \mathbb{E}\left[f(\boldsymbol{\theta}_g^{t+1})\right] + \Phi_c(\eta_c) \\
& + \frac{\eta_s}{2N}\sum_{i=1}^N G_s^2 d_{c,i}^t + \frac{L}{2}N^2\eta_s^2 G_s^2
\end{aligned}
\tag{52}
$$

$$
\begin{aligned}
\Longrightarrow \min\{\frac{\eta_c h}{4}, \frac{\eta_s(2N-1)}{4}\}\|\nabla f(\boldsymbol{\theta}_g^t)\|^2 \leq & f(\boldsymbol{\theta}_g^t) - \mathbb{E}\left[f(\boldsymbol{\theta}_g^{t+1})\right] + \Phi_c(\eta_c) \\
& + \frac{\eta_s}{2N}\sum_{i=1}^N G_s^2 d_{c,i}^t + \frac{L}{2}N^2\eta_s^2 G_s^2
\end{aligned}
\tag{53}
$$

$$
\|\nabla f(\boldsymbol{\theta}_g^t)\|^2 \leq \frac{f(\boldsymbol{\theta}_g^t) - \mathbb{E}\left[f(\boldsymbol{\theta}_g^{t+1})\right] + \Phi_c(\eta_c) + \frac{\eta_s}{2N}\sum_{i=1}^N G_s^2 d_{c,i}^t + \frac{L}{2}N^2\eta_s^2 G_s^2}{\min\{\frac{\eta_c h}{4}, \frac{\eta_s(2N-1)}{4}\}}
\tag{54}
$$

Taking full expectation on both sides, and summing over $t$ from 1 to $T$, we have:

$$
\begin{aligned}
\min_{t\in[T]} \mathbb{E}\left[\|\nabla f(\boldsymbol{\theta}_g^t)\|^2\right] \\
\leq \frac{f(\boldsymbol{\theta}_g^t) - f(\boldsymbol{\theta}_g^*)}{\min\{\frac{\eta_c h}{4}, \frac{\eta_s(2N-1)}{4}\}T} + \frac{\frac{\eta_s}{2N}G_s^2\sum_{i=1}^N d_{c,i}^t}{\min\{\frac{\eta_c h}{4}, \frac{\eta_s(2N-1)}{4}\}} + \frac{\frac{L}{2}N^2\eta_s^2 G_s^2}{\min\{\frac{\eta_c h}{4}, \frac{\eta_s(2N-1)}{4}\}} \\
+ \frac{1}{\min\{\frac{\eta_c h}{4}, \frac{\eta_s(2N-1)}{4}\}}\left[\eta_c^2\left(\frac{6hLdG_c^2}{NB} + \frac{18hL\sigma_c^2}{N}\right) + \eta_c\left(\frac{d^2 L^2 h\mu^2}{48B} + \frac{13hL^2\mu^2}{12}\right)\right]
\end{aligned}
\tag{55}
$$

with respect to $\eta_c \leq \min\{\frac{1}{3Lh}, \frac{2}{NLh^2}, \frac{N}{72L}\}, \forall t \in [T]$.

We want the convergence rate to hold at the same level for both the client-side and server-side, so first we set $\eta = \eta_c h/4 = (2N-1)\eta_s/4$, then we can rewrite the above bound as:

$$
\begin{aligned}
\min_{t\in[T]} \mathbb{E}\left[\|\nabla f(\boldsymbol{\theta}_g^t)\|^2\right] \\
\leq \frac{f(\boldsymbol{\theta}_g^t) - f(\boldsymbol{\theta}_g^*)}{\eta T} + \frac{\frac{\eta_s}{2N}G_s^2\sum_{i=1}^N d_{c,i}^t}{\eta} + \frac{\frac{L}{2}N^2\eta_s^2 G_s^2}{\eta} \\
+ \frac{1}{\eta}\left[\eta_c^2\left(\frac{6hLdG_c^2}{NB} + \frac{18hL\sigma_c^2}{N}\right) + \eta_c\left(\frac{d^2 L^2 h\mu^2}{48B} + \frac{13hL^2\mu^2}{12}\right)\right] \\
= \frac{f(\boldsymbol{\theta}_g^t) - f(\boldsymbol{\theta}_g^*)}{\eta T} + \left(8LG_s^2\frac{N^2}{(2N-1)^2} + 96(\frac{LdG_c^2}{hNB} + \frac{3L\sigma_c^2}{hN})\right)\eta \\
+ \frac{1}{N(2N-1)}G_s^2\sum_{i=1}^N d_{c,i}^t + \left(\frac{d^2 L^2\mu^2}{12B} + \frac{13L^2\mu^2}{3}\right)
\end{aligned}
\tag{56}
$$

then we should have

$$\eta = \mathcal{O}\left(\sqrt{\frac{hNB}{dT}}\right), \eta_c = \mathcal{O}\left(\sqrt{\frac{NB}{dhT}}\right), \eta_s = \mathcal{O}\left(\sqrt{\frac{hB}{dNT}}\right), \tag{57}$$

and $\mu = \mathcal{O}(dhNBT)^{-\frac{1}{4}}$, and we can obtain the convergence rate as:

$$\min_{t \in [T]} \mathbb{E}\left[\|\nabla f(\boldsymbol{\theta}_g^t)\|^2\right] \leq \mathcal{O}\left(\sqrt{\frac{d}{hNBT}}\right) + \mathcal{O}\left(\sqrt{\frac{1}{dhNBT}}\right). \tag{58}$$

Then we complete the proof. $\qquad\square$

### A.4 PROOF OF THEOREM 4.7

In this section, we consider the convergence behavior from the perspective of the language model fine-tuning situation. Since the loss landscape of deep learning lies in a very low-dimensional subspace, where the Hessian of the loss has a remarkably low effective rank, we can leverage this property to analyze the convergence rates more effectively.

**Lemma A.6** (**Client-side bound with low effective-rank**). *Under Assumption 4.1–4.3, and 4.6, drawing $\boldsymbol{u}_i^t$ from the uniform distribution on the unit sphere with radius $\sqrt{d}$, it holds the contribution from the client side:*

$$\begin{aligned}
\mathbb{E}[\mathcal{C}] \leq &-\frac{\eta_c}{4}\|\nabla f_c(\boldsymbol{\theta}_c^t)\|^2 + \frac{\eta_c\mu^2 L^2}{8}(d+3)^3 + \eta_c^2 L^4\mu^2 d^3 \\
&+ \eta_c^2 L\left(1 + \frac{d\kappa + d - 2}{d+2}\right)\frac{1}{N}(\sigma^2 + G_s^2) \\
&+ \eta_c^3 L^2\left(1 + \frac{d\kappa + d - 2}{d+2}\right)^2.
\end{aligned} \tag{59}$$

Since we hold the same assumptions as the proof of Theorem 2 in Li et al. (2024), we use the results of Equation (69) in this paper with characters adapted to our notation, which is given as follows:

$$\begin{aligned}
\mathbb{E}[\mathcal{C}] \leq &-\frac{\eta_c}{2}\|\nabla f_c(\boldsymbol{\theta}_c^t)\|^2 + \frac{\eta_c\mu^2 L^2}{8}(d+3)^3 + \eta_c^2 L^4\mu^2 d^3 \\
&+ \eta_c^2 L\left(1 + \frac{d\kappa + d - 2}{d+2}\right)\left(\|\nabla f_c(\boldsymbol{\theta}_c^t)\| + \frac{1}{N}(\sigma^2 + G_s^2)\right).
\end{aligned} \tag{60}$$

This is achieved by applying Young's inequality. Let us first isolate the terms dependent on the gradient norm from the right-hand side (RHS) of Eq. 60:

$$\text{RHS} \leq -\frac{\eta_c}{2}\|\nabla f_c(\boldsymbol{\theta}_c^t)\|^2 + \eta_c^2 L\left(1 + \frac{d\kappa + d - 2}{d+2}\right)\|\nabla f_c(\boldsymbol{\theta}_c^t)\| + C_1, \tag{61}$$

where $C_1$ collects all terms that are independent of $\|\nabla f_c(\boldsymbol{\theta}_c^t)\|$:

$$C_1 = \frac{\eta_c\mu^2 L^2}{8}(d+3)^3 + \eta_c^2 L^4\mu^2 d^3 + \eta_c^2 L\left(1 + \frac{d\kappa + d - 2}{d+2}\right)\frac{1}{N}(\sigma^2 + G_s^2).$$

We use Young's inequality, which states that for any $a, b \geq 0$ and $\delta > 0$, we have $ab \leq \frac{\delta}{2}a^2 + \frac{1}{2\delta}b^2$. We apply this to the linear gradient term in Eq. 61 by defining:

$$a := \eta_c L\left(1 + \frac{d\kappa + d - 2}{d+2}\right),$$

$$b := \eta_c\|\nabla f_c(\boldsymbol{\theta}_c^t)\|.$$

This application yields the following bound:

$$\begin{aligned}
\eta_c^2 L\left(1 + \frac{d\kappa + d - 2}{d+2}\right)\|\nabla f_c(\boldsymbol{\theta}_c^t)\| &\leq \frac{\delta}{2}\left[\eta_c L\left(1 + \frac{d\kappa + d - 2}{d+2}\right)\right]^2 + \frac{1}{2\delta}\left[\eta_c\|\nabla f_c(\boldsymbol{\theta}_c^t)\|\right]^2 \\
&= \frac{\delta\eta_c^2 L^2}{2}\left(1 + \frac{d\kappa + d - 2}{d+2}\right)^2 + \frac{\eta_c^2}{2\delta}\|\nabla f_c(\boldsymbol{\theta}_c^t)\|^2.
\end{aligned} \tag{62}$$

Substituting the bound Eq. 62 back into our main expression, we can group the coefficients of the $\|\nabla f_c(\boldsymbol{\theta}_c^t)\|^2$ term:

$$\text{RHS} \le \left(-\frac{\eta_c}{2} + \frac{\eta_c^2}{2\delta}\right) \|\nabla f_c(\boldsymbol{\theta}_c^t)\|^2 + C_1 + \frac{\delta\eta_c^2 L^2}{2}\left(1 + \frac{d\kappa + d - 2}{d+2}\right)^2. \tag{63}$$

To simplify the coefficient of the squared gradient norm to a more convenient form, such as $-\frac{\eta_c}{4}$, we select a specific value for the free parameter $\delta$. By setting the new coefficient to this target, we solve for $\delta$:

$$-\frac{\eta_c}{2} + \frac{\eta_c^2}{2\delta} = -\frac{\eta_c}{4}$$

$$\implies \frac{\eta_c^2}{2\delta} = \frac{\eta_c}{2} - \frac{\eta_c}{4} = \frac{\eta_c}{4}$$

$$\implies \delta = 2\eta_c.$$

Since the learning rate $\eta_c > 0$, our choice $\delta > 0$ is valid. With $\delta = 2\eta_c$, the new term arising from Young's inequality that is independent of the gradient becomes $\eta_c^3 L^2 \left(1 + \frac{d\kappa + d - 2}{d+2}\right)^2$. By substituting this result into Eq. 63, we obtain the simplified upper bound for $\mathbb{E}[\mathcal{C}]$. This final form is advantageous for convergence analysis, as the negative squared gradient term is now clearly isolated from other terms that are of a higher order in $\eta_c$ or are related to statistical variance.

### A.4.1 PROOF OF THEOREM 4.7

Following the proof strategy of Theorem 1, the analysis can be naturally divided into two parts: client-side optimization and server-side optimization. Now we are ready to prove Theorem 4.7.

*Proof.* Combining the bounds of $\mathbb{E}_t[\mathcal{C}]$ from Lemma A.6 and $\mathbb{E}_t[\mathcal{S}]$ same as Eq. 49, we have:

$$\begin{aligned}
\mathbb{E}_t[\mathcal{C} + \mathcal{S}] &= \mathbb{E}[\mathcal{C} + \mathcal{S}] \\
&\le -\frac{\eta_c}{4}\|\nabla f_c(\boldsymbol{\theta}_c^t)\|^2 + \Phi_c'(\eta_c) \\
&\quad + \frac{\eta_s G_s^2}{2N}\sum_{i=1}^N d_{c,i}^t - \frac{\eta_s(2N-1)}{4}\|\nabla f_s(\boldsymbol{\theta}_s^t)\|^2 + \frac{L}{2}N^2\eta_s^2 G_s^2
\end{aligned} \tag{64}$$

where $\Phi_c'$ is defined as:

$$\begin{aligned}
\Phi_c' &= \frac{\eta_c\mu^2 L^2}{8}(d+3)^3 + \eta_c^2 L^4 \mu^2 d^3 + \eta_c^2 L\left(1 + \frac{d\kappa + d - 2}{d+2}\right)\frac{1}{N}(\sigma^2 + G_s^2) \\
&\quad + \eta_c^3 L^2\left(1 + \frac{d\kappa + d - 2}{d+2}\right)^2.
\end{aligned} \tag{65}$$

With the same methods in proof of Theorem 4.1, we can have

$$\|\nabla f(\boldsymbol{\theta}_g^t)\|^2 \le \frac{f(\boldsymbol{\theta}_g^t) - \mathbb{E}\left[f(\boldsymbol{\theta}_g^{t+1})\right] + \Phi_c'(\eta_c) + \frac{\eta_s}{2N}\sum_{i=1}^N G_s^2 d_{c,i}^t + \frac{L}{2}N^2\eta_s^2 G_s^2}{\min\{\frac{\eta_c}{4}, \frac{\eta_s(2N-1)}{4}\}} \tag{66}$$

Taking full expectation on both sides, and summing over $t$ from 1 to $T$ (with Assumption 4.4), we have:

$$\begin{aligned}
\min_{t\in[T]}\mathbb{E}&\left[\|\nabla f(\boldsymbol{\theta}_g^t)\|^2\right] \\
&\le \frac{f(\boldsymbol{\theta}_g^1) - f(\boldsymbol{\theta}_g^{T+1})}{\min\{\frac{\eta_c}{4}, \frac{\eta_s(2N-1)}{4}\}T} + \frac{\frac{\eta_s}{2N}G_s^2\sum_{i=1}^N d_{c,i}^t}{\min\{\frac{\eta_c}{4}, \frac{\eta_s(2N-1)}{4}\}} + \frac{\frac{L}{2}N^2\eta_s^2 G_s^2}{\min\{\frac{\eta_c}{4}, \frac{\eta_s(2N-1)}{4}\}} \\
&\quad + \frac{\Phi_c'(\eta_c)}{\min\{\frac{\eta_c}{4}, \frac{\eta_s(2N-1)}{4}\}}.
\end{aligned} \tag{67}$$

We want the convergence rate can hold at the same level for both the client-side and server-side, so first we set $\eta = \eta_c/4 = (2N-1)\eta_s/4$, then we can rewrite the above bound as:

$$
\begin{aligned}
\min_{t\in[T]} &\mathbb{E}\left[\|\nabla f(\boldsymbol{\theta}_g^t)\|^2\right] \\
&\leq \frac{f(\boldsymbol{\theta}_g^1) - f(\boldsymbol{\theta}_g^{T+1})}{\eta T} \\
&\quad + \eta \left[\frac{8LN^2G_s^2}{(2N-1)^2} + 16L^4\mu^2 d^3 + \frac{16L}{N}\left(1 + \frac{d\kappa + d - 2}{d+2}\right)(\sigma^2 + G_s^2)\right] \\
&\quad + \eta^2 \left[64L^2\left(1 + \frac{d\kappa + d - 2}{d+2}\right)^2\right] \\
&\quad + \left[\frac{2G_s^2}{N(2N-1)}\sum_{i=1}^N d_{c,i}^t + \frac{\mu^2 L^2}{2}(d+3)^3\right].
\end{aligned}
\tag{68}
$$

To achieve a more informative rate, we specify the structure of the dominant terms. In many federated learning analyses, the coefficient of the leading $O(\eta)$ error term scales with key system parameters. Let us assume the dominant part of this coefficient is characterized by the condition number $\kappa$, the number of clients $N$, and the average local data size $B$. We can thus define the coefficient of the primary $O(\eta)$ term as being of order $\mathcal{O}(\kappa/(NB))$.

Then we should have the learning rates set by balancing the $O(1/(\eta T))$ and the dominant $O(\eta)$ terms to optimize the bound. This balance, $\frac{1}{\eta T} \approx \eta \frac{\kappa}{NB}$, yields $\eta \propto \sqrt{NB/(\kappa T)}$. We thus set:

$$
\eta = \mathcal{O}\left(\sqrt{\frac{NB}{\kappa T}}\right), \quad \eta_c = \mathcal{O}\left(\sqrt{\frac{NB}{\kappa T}}\right), \quad \eta_s = \mathcal{O}\left(\sqrt{\frac{B}{N\kappa T}}\right),
\tag{69}
$$

and $\mu \leq \frac{\sqrt[4]{\kappa}}{\sqrt[4]{NT}\sqrt{(d+3)^3}}$, we can obtain the convergence rate as:

$$
\min_{t\in[T]} \mathbb{E}\left[\|\nabla f(\boldsymbol{\theta}_g^t)\|^2\right] \leq \mathcal{O}\left(\sqrt{\frac{\kappa}{NBT}}\right) + \mathcal{O}\left(\frac{1}{T}\right) + C_{err},
\tag{70}
$$

where the $O(1/T)$ term arises from the $\eta^2$ components of the bound, and $C_{err} = \frac{2}{\delta}\left[\frac{2G_s^2}{N(2N-1)}\Delta + \frac{\mu^2 L^2}{2}(d+3)^3\right]$ is a constant error floor independent of $T$, indicating convergence to a neighborhood of the optimum.

Then we complete the proof. $\qquad\square$

# B  PSEUDO CODE OF HERON-SFL

---

**Algorithm 1** Hybrid Zeroth- and First-Order Optimization SFL (HERON-SFL)

---

**Require:** Client learning rate $\eta_c$, Server learning rate $\eta_s$, ZO radius $\mu$, local steps $h$, upload period $k$

1: **Server Initialization:**
2: Initialize global model $\boldsymbol{\theta}_g^0 = \{\boldsymbol{\theta}_c^0, \boldsymbol{\theta}_s^0\}$ and auxiliary model $\boldsymbol{\theta}_a^0$
3: Fed-Server broadcasts $\boldsymbol{\theta}_c^0, \boldsymbol{\theta}_a^0$ to all clients
4: **procedure** CLIENTUPDATE$(i, \boldsymbol{\theta}_c^t, \boldsymbol{\theta}_a^t)$
5:      $\boldsymbol{\theta}_{c,i}^{t,0} \leftarrow \boldsymbol{\theta}_c^t, \boldsymbol{\theta}_{a,i}^{t,0} \leftarrow \boldsymbol{\theta}_a^t$
6:      **for** $m = 0, 1, \ldots, h-1$ **do**                     ▷ Local Training Steps
7:          Sample mini-batch $\xi_i$ and random direction $u^{t,m}$
8:          Compute $\Delta\ell_i$ by perturbating $\boldsymbol{\theta}_{l,i}^{t,m} = \{\boldsymbol{\theta}_{c,i}^{t,m}, \boldsymbol{\theta}_{a,i}^{t,m}\}$ with $\pm\mu\boldsymbol{u}^{t,m}$
9:          Estimate ZO gradients: $\hat{\boldsymbol{g}}_{l,i}^{t,m} \leftarrow \hat{\nabla} f_{l,i}(\boldsymbol{\theta}_l; \xi_i))$
10:         Update local models: $\boldsymbol{\theta}_{l,i}^{t,m+1} \leftarrow \boldsymbol{\theta}_{l,i}^{t,m} - \eta_c \hat{\boldsymbol{g}}_{l,i}^{t,m}$
11:         **if** $(m+1) \bmod k = 0$ **then**                     ▷ Periodic upload every $k$ steps
12:             Generate smashed data $\boldsymbol{s}_i^{t,m} \leftarrow \boldsymbol{\theta}_{c,i}^{t,m+1}(\xi_i)$ and send to Main-Server
13:     **return** $\boldsymbol{\theta}_{c,i}^{t,h}, \boldsymbol{\theta}_{a,i}^{t,h}$ to Fed-Server

14: **for** $t = 0, 1, \ldots, T-1$ **do**                     ▷ Main Training Loop (Global Rounds)
15:     *// — Phase 1: Parallel Client Training & Concurrent Server Updates —*
16:     **for** each client $i = 1, \ldots, N$ **in parallel do**
17:         Execute CLIENTUPDATE$(i, \boldsymbol{\theta}_c^t, \boldsymbol{\theta}_a^t)$
18:                                 ▷ Concurrently, server receives periodic client uploads
19:     Main-Server collects all received data $\{\boldsymbol{s}_i^{t,m}\}$ from the round
20:     Main-Server sequentially updates $\boldsymbol{\theta}_s^t$ using all received smashed data: $\boldsymbol{\theta}_s^{t+1} \leftarrow$ updated $\boldsymbol{\theta}_s^t$

21:     *// — Phase 2: Federated Aggregation —*
22:     Fed-Server receives final local models $\{\boldsymbol{\theta}_{c,i}^{t,h}, \boldsymbol{\theta}_{a,i}^{t,h}\}_{i=1}^N$
23:     $\boldsymbol{\theta}_c^{t+1} \leftarrow \frac{1}{N}\sum_{i=1}^N \boldsymbol{\theta}_{c,i}^{t,h}$ and $\boldsymbol{\theta}_a^{t+1} \leftarrow \frac{1}{N}\sum_{i=1}^N \boldsymbol{\theta}_{a,i}^{t,h}$
24:     Fed-Server broadcasts $\boldsymbol{\theta}_c^{t+1}, \boldsymbol{\theta}_a^{t+1}$ for the next round

---

# C  EXPERIMENTAL DETAILS

## C.1  DETAILS OF THE OVERHEAD EVALUATION SETUP

**Communication Cost Measurement.** The client–server communication load is computed by measuring the total number of bits transmitted during each local update step. For smashed-data uploads, we follow the definition in Table 1 and calculate the communication as the tensor size of the smashed activations (i.e., the number of elements multiplied by the batch size) times the numerical precision used in transmission (FP16 in all our experiments). For federated aggregation, we additionally account for the upload of the client-side and auxiliary model parameters, multiplied by the same precision. The overall communication per update step is therefore the sum of (i) smashed-data upload bits and (ii) model-parameter upload bits.

**Client-Side Peak Footprint per Local Update Step.** Peak FP for client-side model update is measured using `torch.cuda.max_memory_allocated()` during a single local client update, capturing the maximum GPU allocation—including **model parameters**, **optimizer states**, **intermediate activations**, and all **temporary CUDA buffers**—required by the update step. For FO baselines (SFLV1/V2, CSE-FSL, FSL-SAGE), the peak occurs during backpropagation because all layerwise activations of both the client model and the auxiliary network must be cached. In contrast, HERON-SFL performs only forward evaluations for its ZO update, so no activations are stored, and the perturbation direction is generated procedurally from a seed rather than stored as a full vector.

**Client-Side Peak FLOPs per Local Update Step.** To measure the peak FLOPs incurred by the client during a single model-update step, we use PyTorch's operator-level FLOP instrumentation, which records the floating-point operations executed by all CUDA kernels involved in forward or backward computations. For first-order baselines (SFLV1/V2, CSE-FSL, FSL-SAGE), we profile the entire forward-backward pipeline of the client-side model and auxiliary network using `torch.profiler.profile()`, and sum all FLOP counts across recorded events. For HERON-SFL, we profile the two forward evaluations required by the two-point ZO estimator, without any backward operations. The peak FLOPs of the client step are computed as the sum of FLOPs from two forward passes. All reported numbers represent FLOPs executed within one local update step on a single client GPU.

## D ADDITIONAL EXPERIMENTS

### D.1 ABLATION STUDY ON HYPER-PARAMETERS OF SFL

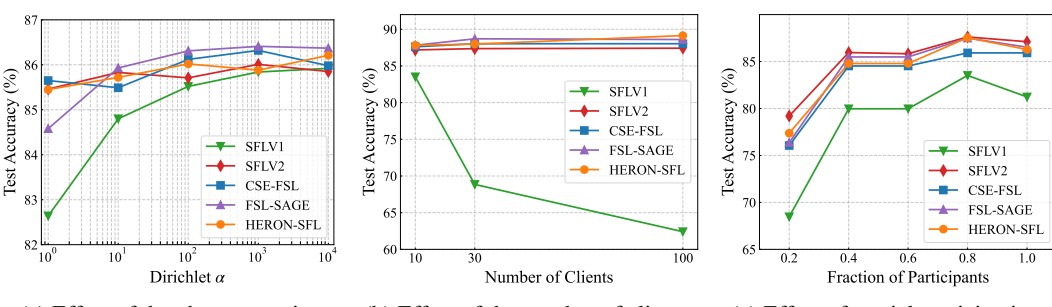

(a) Effect of data heterogeneity.   (b) Effect of the number of clients.   (c) Effect of partial participation.

Figure 5: Test accuracy of different SFL algorithms on CIFAR-10 using a ResNet-18 model. (a) Impact of data heterogeneity under varying Dirichlet $\alpha$ values. (b) Client scalability under different total numbers of clients. (c) Performance under different fractions of participating clients per round.

#### D.1.1 EFFECT OF DATA HETEROGENEITY (NON-IID)

The impact of data heterogeneity is evaluated on CIFAR-10 using a ResNet-18 model with ten clients under full participation. As shown in Figure 5a, varying the Dirichlet concentration parameter creates a broad range of non-IID conditions, yet HERON-SFL maintains accuracy comparable to first-order SFL baselines across all levels of heterogeneity. The zeroth-order updates do not weaken the model's ability to handle distributional shifts, and the perturbation-induced noise remains well controlled. These results indicate that HERON-SFL preserves the robustness to non-IID client data that is characteristic of first-order SFL training.

Additionally, we further test the training accuracy on $\alpha = 0.1$, an extremely heterogeneous setting where client label distributions exhibit minimal overlap. Under this regime, all SFL variants fail to converge, which aligns with findings in prior FL studies where such severe non-IID conditions cause strong client drift and unstable global updates. Achieving stable training at $\alpha = 0.1$ typically requires dedicated mechanisms such as data sharing (Zhu et al., 2021), gradient regularization (Li et al., 2020), or distribution alignment (Mahmud & Dividino, 2024), none of which are incorporated in standard SFL pipelines. A full investigation of these techniques falls outside the scope of this work, and our study therefore focuses on the heterogeneity regimes commonly examined in the SFL work (Nair et al., 2025). We emphasize that this failure mode is not caused by the use of zeroth-order optimization: even first-order SFL baselines collapse under such extreme non-IID conditions, consistent with prior FL findings.

#### D.1.2 EFFECT OF THE NUMBER OF CLIENTS

Scalability is examined by varying the total number of clients while keeping the dataset (CIFAR-10), model architecture (ResNet-18), and full participation unchanged under an IID configuration. As shown in Figure 5b, HERON-SFL sustains nearly identical accuracy as the federation expands from ten to one hundred clients, demonstrating that our HERON-SFL remains stable at larger scales.

### D.1.3 EFFECT OF PARTIAL PARTICIPATION

Training performance under partial participation is assessed on CIFAR-10 with a ResNet-18 model and 10 IID clients. As shown in Figure 5c, HERON-SFL maintains stable accuracy over a wide range of participation ratios, including cases where only a small fraction of clients contributes updates in each round. Its behavior closely matches that of first-order SFL baselines, indicating that partial participation does not impair the convergence behavior of the zeroth-order client updates. These findings confirm that HERON-SFL remains reliable even when participation is limited, a setting commonly encountered in practical cross-device federated learning.

### D.2 ABLATION STUDY ON HYPER-PARAMETERS OF ZO

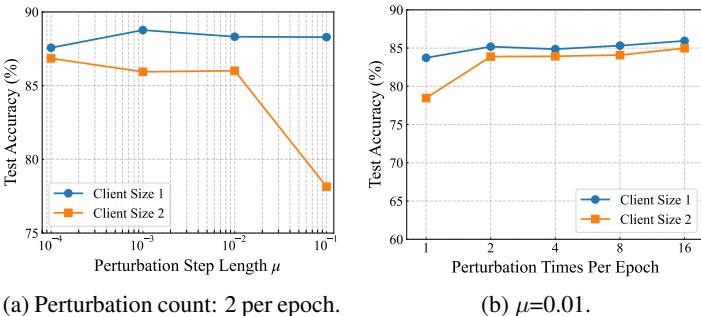

(a) Perturbation count: 2 per epoch.           (b) $\mu$=0.01.

Figure 6: Ablation study on local zeroth-order training hyper-parameters using ResNet-18 on CIFAR-10 under an IID setting with ten clients, with all experiments using the same auxiliary model implemented as a single linear layer. Client Size 1 denotes the first convolutional layer and one residual block on the client, while Client Size 2 places three residual blocks on the client. (a) Test accuracy under different perturbation step lengths $\mu$. (b) Test accuracy under different perturbation counts per epoch.

As shown in Figure 6, HERON-SFL exhibits stable performance across a wide range of zeroth-order hyper-parameters, demonstrating robustness to both the perturbation step size and the number of perturbations per epoch. When an appropriate step size $\mu$ is selected, using only two perturbations per epoch is sufficient to ensure reliable convergence, indicating that HERON-SFL does not suffer from the instability often associated with zeroth-order optimization. Across both figures, Client Size 1 consistently achieves higher accuracy than Client Size 2 (align with the ablation experiments in Figure 4), reflecting the expected increase in optimization difficulty when a larger portion of the model is placed on the client. This mild degradation is acceptable in SFL settings because resource-constrained devices typically hold only small client sub-models, while the majority of parameters remain on the server for first-order training. Overall, the results confirm that HERON-SFL maintains strong accuracy under practical ZO configurations and remains reliable even when client-side capacity varies.

### D.3 EVIDENCE FOR LOW RANK ASSUMPTION

Given the prohibitive cost of full Hessian computation for LLMs, we validated the low-effective rank assumption using a modified ResNet-18 (He et al., 2016) on CIFAR-10. We estimated the Hessian eigenvalue density via the stochastic Lanczos algorithm (Golub & Welsch, 1969), following the methodology of Ghorbani et al. (2019). As shown in Figure 7, the resulting distribution, heavily concentrated at zero, suggests that the low-rank structure is an intrinsic property of the optimization landscape rather than a strong constraint. For empirical evidence of the low-rank assumption, the same evidence can be seen in Li et al. (2024) Appendix C.3.1.

This observation extends to the regime of LLMs, particularly during the fine-tuning phase. Recent works, such as GaLore (Zhao et al., 2024), have provided robust evidence that while pre-training may necessitate high-rank updates, the weight modifications required for fine-tuning naturally reside in a low-rank subspace. This intrinsic low-dimensionality is a critical factor explaining the success of Zeroth-Order (ZO) optimization methods in this domain. It theoretically justifies why methods like

MeZO (Malladi et al., 2023) can achieve competitive performance with memory-efficient, gradient-free updates, as they effectively navigate this low-rank manifold.

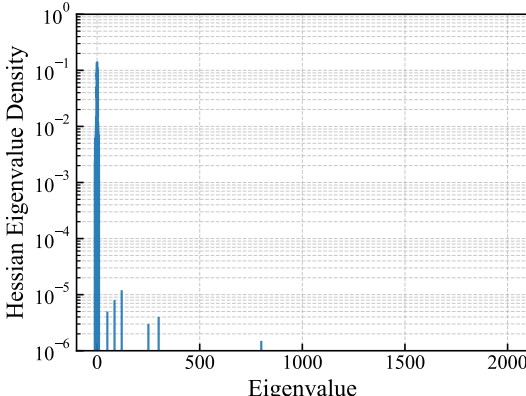

Figure 7: Hessian eigenvalue distribution with training custom ResNet on CIFAR-10 dataset.

## E    LLM USAGE STATEMENT

We acknowledge the use of a Large Language Model as a general-purpose assist tool in preparing this paper. The LLM was used only for language assistance, including polishing grammar, improving clarity, and refining the flow of the text.

The research ideas, experiments, analyses, and conclusions presented in this work are solely the result of the authors' efforts. The LLM did not contribute to the design of experiments, development of algorithms, data analysis, or any substantive scientific content.

