# OpenReview forum: "Lean Clients, Full Accuracy: Hybrid Zeroth- and First-Order Split Federated Learning"
_ICLR.cc/2026/Conference — ICLR 2026 Conference Withdrawn Submission_

### Official Review · Reviewer_7LFG · 2025-10-26

**Soundness:** 3
**Presentation:** 3
**Contribution:** 2
**Rating:** 2
**Confidence:** 5

**Summary:**

The paper introduces a framework that enhances Split Federated Learning (SFL) by combining zeroth-order (ZO) and first-order (FO) optimization methods. Traditional SFL suffers from high computational and memory demands on client devices due to backpropagation. HERON-SFL addresses this by using lightweight ZO optimization on the client side, while retaining FO optimization on the server. This hybrid approach eliminates costly backpropagation on edge devices, substantially reducing client-side memory and computation without sacrificing model accuracy. Theoretical analysis shows that HERON-SFL achieves convergence rates independent of model dimensionality under a low effective-rank assumption. Experiments on ResNet training and GPT-2 fine-tuning tasks confirm that HERON-SFL matches benchmark accuracy while greatly improving efficiency, enabling large-scale model training on resource-constrained devices.

**Strengths:**

+ Introduces HERON-SFL, the hybrid zeroth-order (ZO) and first-order (FO) optimization framework for Split Federated Learning (SFL). This hybridization smartly leverages ZO for clients (forward-only computation) and FO for servers (precise gradient updates), balancing computational efficiency and accuracy.

+ Provides a rigorous convergence analysis for HERON-SFL. Shows that under a low effective-rank assumption, the convergence rate becomes independent of model dimensionality, overcoming a major limitation of traditional ZO methods.

+ Validated on both vision (ResNet on CIFAR-10) and language (GPT-2 fine-tuning on E2E dataset) tasks. Matches or surpasses the accuracy of SFL baselines (e.g., FSL-SAGE, CSE-FSL) with substantially lower resource use.

**Weaknesses:**

- The experiments focus on ResNet-18 (CIFAR-10) and GPT-2 (E2E dataset) — both relatively moderate-scale tasks. There is no evaluation on truly large foundation models (e.g., GPT-3-scale or ViT-level networks) where the claimed scalability advantages would be more convincing.
- Client-device heterogeneity (e.g., varying compute or network speeds) is not experimentally explored, which is critical in federated settings.
- Although communication costs are analyzed, real-world communication latency, bandwidth constraints, and asynchrony are not simulated. No analysis of packet loss, network delay, or intermittent connectivity, which are common in federated edge systems.
- While the paper claims that ZO variance is mitigated by server-side FO refinement, it lacks a quantitative analysis of gradient noise or bias across training. ZO methods are inherently noisy and can be sensitive to perturbation size, but the paper does not offer sensitivity or ablation studies on this hyperparameter. Also, from Fig. 2, the convergence curve of HERON-SFL under non-IID case is not stable, indicating that ZO variance still has negative impact on the convergence of HERON-SFL. High-variance updates could lead to instability, particularly in non-IID or large-scale client distributions, but this is not deeply examined.
- Theoretical convergence independence from model dimensionality hinges on the “low effective rank” assumption, which may not hold universally — especially in deep or overparameterized models. The paper does not empirically validate whether this assumption is satisfied in practice for models like GPT-2 or ResNet-18.
- The paper does not compare with recent SFL paper [R1], which proposes a dynamic tiering approach for SFL to address the computation and communication challenges in SFL. Also, the paper compares only with SFL-based baselines, and comparisons with other related FL methods such as those cited in the paper are not provided.

[R1] Mohammadabadi, Seyed Mahmoud Sajjadi, Syed Zawad, Feng Yan, and Lei Yang. "Speed up federated learning in heterogeneous environments: a dynamic tiering approach." IEEE Internet of Things Journal (2024).

- The paper does not clearly specify client compute capabilities, memory constraints, or network parameters, making it difficult to gauge real-world feasibility. This lack of transparency weakens the empirical section’s reproducibility and external validity.
- Although some non-IID experiments are reported, there is no study on extreme heterogeneity, client dropouts, or adversarial behaviors in real SFL deployments. Security and privacy implications of zeroth-order updates (which could leak model outputs) are not discussed.

**Questions:**

1. Can HERON-SFL scale to large models like GPT-3 or ViT while maintaining efficiency?
2. How does HERON-SFL handle clients with different compute power or network speeds?
3. How does the system perform under real-world communication network (with latency, bandwidth limits, packet loss)?
4. What quantitative evidence supports that FO refinement reduces ZO variance? How sensitive is HERON-SFL to perturbation size ($\mu$)?
5. The experiments use only 3 to 5 clients, which is far smaller than typical federated setups involving tens or hundreds of participants. How does HERON-SFL scale as the number of clients increases?
6. How does HERON-SFL perform under various non-IID, dropout, or adversarial conditions? Could ZO updates leak information through activation outputs?

---

> ### Author Response · Authors · 2025-11-21
>
> > Weakness 1 & Question 1: evaluation on truly large foundation models
>
> Thank you for raising this point. Our core contribution lies in improving client-side resource efficiency in SFL systems, and the experimental settings reflect the typical scales used in prior SFL works due to device constraints. We believe the proposed hybrid ZO–FO design can naturally extend to larger models, as recent results such as MeZO [1] have already demonstrated the effectiveness and stability of zeroth-order fine-tuning on OPT-66B. Unfortunately, running experiments on GPT-3– or ViT-scale models is beyond our current computational budget, but our theoretical analysis and existing evidence from large-model ZO literature [1] suggest that the scalability benefits of HERON-SFL should carry over to larger models.
>
> [1] Malladi, Sadhika, et al. "Fine-tuning language models with just forward passes."(NeurIPS 2023)
>
> > Weakness 2,3 & Question 2, 3: Real-world consideration
>
> Thank you for highlighting this important point. We agree that real-world factors such as varying compute or network speeds, network latency, bandwidth variability, packet loss, and intermittent connectivity are critical in practical federated deployments. However, these system-level parameters are orthogonal to the core contribution of HERON-SFL, which focuses on reducing per-client memory and computation through hybrid ZO–FO optimization under a standard synchronous SFL setting. A full exploration of heterogeneous communication environments would substantially expand the scope of the paper but not the main focus of our work, and we consider it a valuable direction for future work.
>
> > Weakness 4 & Question 4: ZO Variance/Sensitivity, robustness on non-iid
> * For the ZO hyperparameter ablation,  please see our responses to `Weakness 3 by EnDt`.
> * For non-IID experiments. The slight instability observed in Fig. 2 is primarily due to the non-IID data distribution rather than to zeroth-order updates themselves. We note that the same fluctuation pattern also appears in all compared SFL baselines (CSE-SFL, FSL-SAGE, and SFLV2), indicating that non-IID heterogeneity is a **shared challenge for split federated learning**. To address this more systematically, we have added ablation studies (see Appendix D.1.1) in the revision covering different Dirichlet $\alpha$ values, which confirm that HERON-SFL behaves similarly to other SFL methods as non-IIDness increases. Extreme non-IID scenarios remain difficult for all existing SFL approaches; our work focuses on improving client-side efficiency, and advanced optimization for heterogeneous regimes is beyond the scope of this paper.
>
>
> > Weakness 5: Low rank assumption
>
> Please see our response to `Weakness 4 by FesG`.
>
> > Weakness 6: Compare with DTFL [1]
>
> Thank you for pointing this out. DTFL [1] requires per-round dynamic tier assignments and multi-tier model splits, which fundamentally change the model architecture and training pipeline. In contrast, HERON-SFL keeps the model split and communication pattern identical to CSE-FSL [2]. Because DTFL is a system-level redesign rather than an optimizer choice, comparing them would mix two orthogonal dimensions and lead to uninterpretable results. We appreciate the suggestion, and we will cite DTFL in the related-work section and briefly discuss this distinction.
>
> [1] Mohammadabadi, Seyed Mahmoud Sajjadi, Syed Zawad, Feng Yan, and Lei Yang. "Speed up federated learning in heterogeneous environments: a dynamic tiering approach." (IEEE Internet of Things Journal 2024).
> [2] Mu, Yujia, and Cong Shen. "Communication and storage efficient federated split learning." ICC 2023-IEEE International Conference on Communications. IEEE, 2023.

---

> > ### Author Response · Authors · 2025-11-21
> >
> > > Weakness 7: Real-world considerations;
> >
> > Thank you for raising this point. We fully agree that real-world considerations such as precise client compute capacity, memory budgets, and network configurations are important for system-level SFL deployments. However, addressing these system-level factors requires a dedicated evaluation of hardware heterogeneity, network variability, and deployment constraints, which would substantially expand the methodological scope of the paper. Our focus in this work is to establish the algorithmic feasibility and client-side efficiency of hybrid ZO–FO optimization under a standard synchronous SFL setup.
> >
> >
> >
> > > Weakness 8 & Question 6: non-IID, partial participation, security, and privacy
> >
> > * Thank you for pointing this out. We have added additional ablation non-IID experiments (See Appendix D.1.1).
> > * For the partial participation, we add extra experiments in the revision (see Appendix D.1.3). Experiments show that HERON-SFL can maintain performance comparable to other first-order optimization baselines under different fraction scenarios.
> > * For the problem of security and privacy, please see our response to `Concern 2 by n2hD`.
> >
> >
> > > Question 5: Scalability on the number of clients
> >
> > We have included additional experiments in the revised version (see Appendix D.1.2) demonstrating that HERON-SFL maintains stable performance when the number of participating devices increases, confirming that the proposed method extends naturally to larger-client SFL deployments.

---

> > > ### Comment · Reviewer_7LFG · 2025-11-24
> > >
> > > Thank you to the authors for providing responses to my comments. I will summarize all of my comments below.
> > >
> > > 1. Heterogeneity and real-world conditions:
> > > The authors state that system-level variability is “orthogonal,” but in federated and split learning, heterogeneous compute/network speeds, bandwidth limits, latency, packet loss, and intermittent connectivity directly impact algorithm behavior. Since HERON-SFL’s design affects communication frequency and ZO/FO update interaction, ignoring these factors limits the paper’s practical relevance.
> > >
> > > 2. Comparison with DTFL and other baselines:
> > > The justification that DTFL is a “system-level redesign” does not fully address the concern. DTFL specifically targets SFL efficiency and heterogeneity, which overlaps with the motivations of HERON-SFL. A comparison—even a limited one—would clarify relative advantages.
> > >
> > > 3. ZO variance and sensitivity:
> > > The response does not provide quantitative evidence that FO refinement reduces ZO variance, nor does it include perturbation-size sensitivity studies. Non-IID instability is attributed to data heterogeneity, yet the variance characteristics of ZO under different perturbation scales, noise levels, or model partitions remain unexamined.
> > >
> > > 4. Real-world feasibility:
> > > Client compute/memory/network configurations remain unspecified. Without concrete resource profiles, it is difficult to assess reproducibility or judge whether the proposed method is feasible on realistic edge devices.
> > >
> > > Overall, while I appreciate the updates, the rebuttal does not sufficiently resolve key concerns about heterogeneity, ZO-variance behavior, and the practicality of deploying HERON-SFL in realistic federated environments. My assessment therefore remains largely unchanged.

---

> > > > ### Author Response · Authors · 2025-12-02
> > > >
> > > > Thank you for the follow-up questions.
> > > >
> > > > **Heterogeneity and real-world conditions.**
> > > > We agree that heterogeneity and real-world deployment factors are important in practice. However, we would like to note that our work is scoped as an **algorithmic contribution to SFL** under the *standard synchronous training regime* used in prior SFL papers. Within this scope, we have already included non-IID, partial-participation, and client-count ablations. We therefore do not view the absence of full system-level simulations as a methodological flaw.
> > > >
> > > > **Comparison with DTFL and other baselines.**
> > > > DTFL is designed for *heterogeneous-tier SFL* and relies on a fundamentally different system architecture and set of objectives. In contrast, HERON-SFL operates in a widely studied *fixed-architecture* SFL setting ([1-3]) and introduces innovations specifically at the **optimizer level**. A direct comparison would conflate architectural and algorithmic differences, making it difficult to draw fair or interpretable conclusions about our method’s contributions.
> > > >
> > > >
> > > >
> > > > **ZO variance and sensitivity.**
> > > > We would like to clarify that we do not claim that FO “refines” or stabilizes ZO variance; we have revised the wording in the paper to avoid this misunderstanding. In the updated revision, we also added explicit perturbation-size and probe-count ablations (Appendix D.2), showing that HERON-SFL maintains stable behavior across a range of hyperparameters. The non-IID fluctuations also appear in all SFL baselines and are not specific to ZO in HERON-SFL.
> > > >
> > > > **Real-world feasibility.**
> > > > We have included in concrete FLOPs and peak-memory measurements that characterize client resource requirements in Tables 2–3, which we consider the most relevant feasibility indicators for an *algorithmic SFL* paper. Hardware- and network-specific latency or bandwidth modeling, while valuable, is deployment-dependent and typically addressed in system-level studies. Regarding reproducibility, our uploaded code **fully reproduces** the algorithmic results reported in the paper.
> > > >
> > > > [1] Han, Dong-Jun et al. "Accelerating Federated Learning with Split Learning on Locally Generated Losses." (2021).
> > > >
> > > > [2] Mu et al. "Communication and storage efficient federated split learning." ICC 2023-IEEE International Conference on Communications. IEEE, 2023.
> > > >
> > > > [3] FSL-SAGE: Accelerating Federated Split Learning via Smashed Activation Gradient Estimation (ICML 2025).

---

### Official Review · Reviewer_22nq · 2025-10-28

**Soundness:** 3
**Presentation:** 3
**Contribution:** 2
**Rating:** 6
**Confidence:** 3

**Summary:**

This paper proposes a hybrid approach for split federated learning, in which the server performs first-order optimization while the clients employ zeroth-order optimization to reduce peak memory usage and computational cost. The paper theoretically proves the convergence of the proposed method and empirically demonstrates that it achieves comparable performance to the baselines while offering significant savings in peak memory usage and computational cost.

**Strengths:**

* The idea of reducing memory and computation costs through zeroth-order optimization on the client side, while compensating for it with precise first-order optimization on the server side, is convincing.
* The paper provides evidence, both experimental and theoretical, that the proposed hybrid approach can effectively reduce memory and computation costs without causing performance degradation.
* The paper is well written and easy to follow.

**Weaknesses:**

**The importance of client-side training.**  I believe that the main reason the proposed hybrid optimization does not lead to a notable performance drop is that most of the learning still happens on the server using first-order optimization, while the client side primarily serves to “smash” the data for privacy. Because of this, it is difficult to clearly separate whether the benefit comes from the effectiveness of zeroth-order optimization itself, or simply from the fact that training the client-side model is not that important. For example, it would be helpful to include a comparison experiment between (i) freezing the client-side (encoder) part of a pretrained model and training only the server side, and (ii) applying zeroth-order optimization on the client side. This would clarify the role and necessity of client-side optimization.

**Communication may still remain the primary bottleneck.**  Based on the comparison of communication cost and FLOPs in Table 2, it appears that the main bottleneck lies in communication rather than computation. Although zeroth-order optimization indeed reduces the burden of client-side updates, it is questionable whether client-side updates can truly be considered the bottleneck in the context of split federated learning.

**The experimental setting is too limited.**  Although the paper aims to reduce memory and computation costs for resource-constrained edge devices, using only 3 or 5 clients in a cross-silo setting is insufficient to validate this objective. Please refer to the detailed questions provided in the Questions below.

**Questions:**

* Would there be any difference in performance when the number of clients increases, each holding a smaller amount of data, or when the batch size becomes smaller?
* What is the effect of partial participation compared to full participation?
* What happens when the data distribution becomes more non-IID, for example, when the α value in the Dirichlet distribution is reduced below 1?
* If the client-side model becomes larger, would the disadvantages of zeroth-order optimization become more significant?
* As I understand it, the performance of first-order optimization should serve as the upper bound for that of zeroth-order optimization. However, in Figure 3 and similar results, zeroth-order optimization sometimes performs even better. Why might that be the case?

---

> ### Author Response · Authors · 2025-11-21
>
> We really appreciate your positive and instructive feedback and will provide a piecewise response below.
>
> > Weakness 1: The importance of client-side training.
>
> Thank you for raising this important point. We fully acknowledge that the purpose of SFL is to leverage the server’s strong computational capability for global model training. The reason HERON-SFL does not suffer accuracy degradation is not because the client-side model is unimportant, but because the shallow encoder partition used in SFL naturally satisfies several conditions under which zeroth-order optimization is highly effective. Specifically:
> (1) Shallow layers generally exhibit a flatter loss landscape and are easier to optimize, as observed in a prior study [1].
> (2) SFL inherently assigns only a small and lightweight portion of the network to the clients, making ZO’s variance manageable.
> (3) The auxiliary network decouples client and server updates, enabling stable local training despite using ZO.
>
> [1] Chen, Yixiong et al. "Which layer is learning faster? A systematic exploration of layer-wise convergence rate for deep neural networks."(ICLR, 2023)
>
> Regarding your suggested experiments:
> * We conducted a warm-up stage where clients jointly pretrain the client–auxiliary model pair (similar to FL) before SFL begins. We observed that once server-side training starts, only a few communication rounds are required to reach high global accuracy (e.g., 5 rounds to 80% test accuracy). This confirms that the client-side model indeed needs to reach a reasonably stable parameter state for effective SFL training.
> * We also explored the use of ZO on the server side (we assume this is what the reviewer intended). As expected, the high-variance gradient estimates of ZO led to significantly poorer convergence (only 45.32% test accuracy under the same ResNet-18 CIFAR-10 setting), indicating that **first-order optimization is essential for training the server-side global model**.
>
> > Weakness 2: Communication may still remain the primary bottleneck.
>
> We agree that communication is a major bottleneck in SFL systems. Our baseline methods, such as CSE-SFL and FSL-SAGE reduce communication by roughly half through local-loss training, and HERON-SFL follows the same principle. Building on this reduced-communication regime, our work targets a different but important bottleneck: **client-side training cost**. The key insight is that by removing activation caching and backpropagation, HERON-SFL lowers client-side compute and memory to inference-level complexity, which greatly expands the practicality of SFL on highly resource-constrained edge devices. This enables many devices that could not previously participate in training to contribute their local data, improving scalability in real-world deployments.
>
>
> > Weakness 3 & Question 1, 2, 3: The experimental setting is too limited.
>
> * Thank you for pointing this out. We have added additional ablation experiments (See Appendix D.1) in the revision to more thoroughly evaluate the experimental setting, including varying the **number of clients**, enabling **partial participation**, and testing under different levels of **non-IID data heterogeneity**. These results consistently show that HERON-SFL maintains stable performance while preserving its client-side efficiency advantages.
> * For the **batch-size ablation**, we observe that smaller batches converge more smoothly when using the same learning rate. This is consistent with well-established training dynamics in stochastic optimization, where smaller batches produce lower-variance gradient estimates that often improve stability in the early stages of training.

---

> > ### Author Response · Authors · 2025-11-21
> >
> > > Question 4: Effect of client-side model size;
> >
> >
> > We note that Figure 4 already compares performance under different client model sizes. When the client model is small, such as using three blocks, both HERON-SFL and the first-order baseline CSE-FSL achieve lower training loss. This occurs because a smaller client partition implies a larger server-side model, making representation alignment easier during training.
> > Our zeroth-order ablation study (see Appendix D.2) further confirms that smaller client partitions yield higher test accuracy across settings. This observation is consistent with the nature of resource-constrained SFL settings: client devices are resource-constrained, and there is little incentive to enlarge the client-side model or to employ heavier auxiliary networks. Addressing privacy considerations in this context is important, and we regard this as an interesting direction for future work.
> >
> > > Question 5:  HERON-SFL performs better than FO-based methods in LLM fine-tuning
> >
> > Thank you for the question. The phenomenon in Figure 3, where zeroth-order optimization slightly outperforms first-order optimization, can be attributed to the fact that the low-rank assumption holds particularly well in LLM fine-tuning. This behavior has also been observed and rigorously evaluated in MeZO [1], where ZO-based fine-tuning achieves accuracy matching or even surpassing that of first-order methods under comparable settings. Our results are consistent with these findings. We have added an explanation of this in the revision.
> >
> > [1] Malladi, Sadhika, et al. "Fine-tuning language models with just forward passes."(NeurIPS 2023)

---

### Official Review · Reviewer_FesG · 2025-10-28

**Soundness:** 3
**Presentation:** 3
**Contribution:** 3
**Rating:** 4
**Confidence:** 1

**Summary:**

This paper tackles the issue of limited system resources at the client-side by developing a new split learning + federated learning framework which combines first-order and zeroth-order optimizations. Specifically, the clients update local models using zeroth-order method while the server updates the assigned output-side layers using the first-order method. In addition, the authors employ auxiliary models at the client-side so that the local models can proceed to the backpropagation without waiting for the server to propagate the error back to the individual clients. The convergence properties were analyzed based on relatively strong assumptions. Then, extensive empirical study demonstrates its superior FLOPS-to-target accuracy compared to other FSL methods.

**Strengths:**

1. Theoretical analysis covers iid and non-iid together and also consider the specific split + federated learning settings. This thorough analysis framework will provide readership with a useful starting point of analyzing other algorithms.

2. Client-side resource consumptions are thoroughly analyzed (section 4.2), which clearly shows the benefits from the proposed method.

3. Experiments are fairly extensive covering training from scratch as well as fine-tuning.

**Weaknesses:**

While I see mostly valueable contributions, still I find some limitations as follows.

1. [**Relative performance gain against conventional FL and conventional SplitLearning**] While the theoretical analysis and empirical results demonstrate the efficacy of the proposed method, I am not sure whether it consumes less resources than conventional FL to achieve the same target accuracy. What if the zeroth-order method is applied to a few input-side layers and the first-order method to the rest of the layers? I recommend more thoroughly justifying the proposed framework because it is basically a combination of two existing distributed learning paradigms.

2. [**Latency analysis**] In general, split learning methods suffer from frequent communications (latency cost). Even though the proposed method has fewer communications than the typical split learning methods, thanks to the combination of FL and the proposed auxiliary model, I guess it will still have some latency cost issues. Table 2 only considers bandwidth consumptions. I recommend providing an additional analysis of communication counts.

3. [**Unrealistic number of clients**] The authors use 3~5 clients in the experiments. I believe this is seriously misaligned with the concept of federated learning which is a large-scale distributed learning paradigm. Most federated learning studies use at least 32$\sim$64 clients and many papers use even larger numbers of clients like 128 to 1024. Only with 3 or 5 clients, the whole empirical results do not well support the efficacy of the proposed method. Will the proposed method work well with 128 clients?

4. [**Strong assumptions**] I appreciate the extensive and thorough theoretical analysis provided in Section 4. However, the assumptions are relatively stronger than recent FL studies. E.g., many studies rely on either assumption 2 or 3, not on both of them. In addition, assumption 6 implicitly indicates that the high effective rank of model parameters potentially harms the convergence rate. This is a worrisome property because the higher the effective rank, the stronger the representation capacity of the model. Could authors provide more detailed discussions regarding the meaning of this assumption?

Overall, I think this study provides meaningful and useful insights to Split Learning or Federated Learning researchers. However, due to the above critical limitations, I cannot give a positive score for now. I will re-evaluate the paper after the authors' rebuttal.

**Questions:**

My questions are provided in the above weakness section. Please carefully address them.

---

> ### Author Response · Authors · 2025-11-21
>
> Thank you for your comments and suggestions.
>
> > Weakness 1: Relative performance gain against conventional FL and conventional SplitLearning;
>
> Thank you for pointing this out.
> * HERON-SFL does not aim to replace FL; rather, it addresses the client-side memory bottleneck that prevents FL from training large models on edge devices. Unlike FL or a “partial ZO patch,” only SFL exposes a structural backward dependency that ZO can eliminate. Our hybrid ZO–FO design removes all activation caching on clients—something neither FL nor FO-based SFL can accomplish—resulting in $\mathcal O(1)$ activation memory and enabling substantially larger client-side partitions.
> * It is also important to emphasize that applying zeroth-order updates to the first few layers within a standard federated learning (FL) pipeline does **not** provide the benefits achieved in HERON-SFL. In FL, clients must still run full forward–backward passes for all layers they hold; replacing the first layers with ZO would **not reduce activation memory**, because backpropagation must still be performed for the remaining layers. Most importantly, in FL, this modification **does not reduce communication**, because clients still need to transmit full gradients or model updates every round.
> * In contrast, HERON-SFL uses ZO precisely on the split boundary of an SFL architecture where clients never receive backward gradients; therefore, ZO removes backprop entirely on the client, eliminates activation caching, and reduces both computation and peak memory while maintaining one-round communication. This property is unique to the SFL setting and cannot be achieved by naively inserting ZO into an FL pipeline.
>
> > Weakness 2: Latency analysis;
>
> Thank you for highlighting the importance of analyzing communication latency.
>
> HERON-SFL reduces the number of communication rounds required for each local update. Classical split federated learning needs one forward transmission and one backward transmission for every mini-batch, resulting in two rounds of communication. Local-loss-based SFL (CSE-SFL, FSL-SAGE) reduces this to a single round. HERON-SFL follows the same one-round pattern, and local loss-based updates do not require gradients from the server.
>
> | Method                         | # Forward Transmissions per Step | # Backward Transmissions per Step | Communication Time per Global Model Update |
> |-------------------------------|----------------------------------|-----------------------------------|--------------------------------|
> | SFLV2     | 1                                | 1                                 | **2**                          |
> | FSL-SAGE / CSE-FSL            | 1                                | 0                                 | **1**                          |
> | **HERON-SFL (ours)**          | 1                                | 0  | **1**                          |
>
> As already shown in Figure 2 of the main paper, we provide training accuracy as a function of communication rounds, where one round corresponds to completing one global model update. The results show that HERON-SFL reaches the target accuracy within the same number of model-update rounds. Compared with classical split federated learning, our method (as well as FSL-SAGE and CSE-FSL) requires no backward transmission, thereby reducing the communication cost by roughly half and eliminating the associated backward-pass latency. This directly contributes to lower end-to-end delay per update while preserving accuracy.
>
> > Weakness 3: Unrealistic number of clients;
>
> We clarify that SFL is typically evaluated in the cross-silo setting or edge-devices in a local area, where 3–10 clients is standard practice (as in FSL-SAGE and CSE-FSL). The core contribution of HERON-SFL is reducing per-client resource usage, which scales independently of the number of clients. To further verify scalability, we include additional experiments in the revision (see Appendix D.1.2) showing that HERON-SFL exhibits the **same client-number scaling behavior** as existing FO-based SFL methods.
>
> > Weakness 4: Strong assumptions;
>
> Thank you for pointing this out. The low effective rank assumption is widely observed in modern deep networks, where the Hessian spectrum is dominated by a small number of large eigenvalues and the remaining mass decays rapidly. Our analysis requires only this empirical spectral concentration, not an exact low-rank structure, and prior work [1, 2] has shown that such behavior becomes even more pronounced in CNNs and Transformers. These observations suggest that **the assumption is mild and generally satisfied in practice**. We have added experimental validation and discussion in the revision (see Appendix D.3).
>
> [1] Behrooz Ghorbani, et al. An investigation into neural net optimization
> via Hessian eigenvalue density. (ICML 2019)
> [2] Jiawei Zhao, et al. Galore: Memory-efficient llm training by gradient low-rank projection. (ICML 2024)

---

> > ### Comment · Reviewer_FesG · 2025-11-23
> > **Response to the rebuttal**
> >
> > First, thank you for the rebuttal. Unfortunately, most of my concerns were not addressed.
> >
> > Weakness 1. The relative performance gain against vanilla FL or SL should be further thoroughly analyzed to argue that the proposed method has its own special benefits. The authors said that "ZO–FO design removes all activation caching on clients—something neither FL nor FO-based SFL can accomplish". However, this benefit stems from not only the proposed method but also the fundamental structure of split federated learning (SFL). I still do not see any meaningful comparisons in terms of the computational or communication costs. I would recommend numerically and directly compare the four representative schemes, FL, SL, SFL, and the proposed method.
> >
> > Weakness 2. The provided analysis just counts the number of communications per round and compare the count across different SFL methods. If the latency is very expensive, FL may run more comm rounds within a fixed amount of time, and it results in achieving higher accuracy. Even if it outperforms the existing similar cross-silo SFL methods, it will not mean that the proposed hybrid scheme is a meaningful advancement if it is slower than other popular distributed learning schemes such as FL or SL. I recommend fixing a certain latency cost and numerically analyze and compare the overall training time across several representative distributed learning schemes.
> >
> > Weakness 3. I see the point. Now I have no concerns regarding the number of clients. However, if this work is a cross-silo specific study, the resource-efficiency may not be a significant issue because the cross-silo environments are not typically implemented using resource-limited edge devices.
> >
> > Weakness 4. The heavy-tailed singular value spectrum cannot be just "assumed". It is an **observation** and I can easily produce a well-trained neural network which does not have such distributions by applying a soft orthogonality regularizer. Therefore, I do not agree that the assumption is mild.
> >
> > Overall, the authors' rebuttal have not addressed my concerns well, and I will keep my score negative.

---

> > > ### Author Response · Authors · 2025-12-02
> > >
> > > 1. We would like to clarify that our contribution is explicitly scoped within the standard SFL framework. HERON-SFL leverages the key architectural property of SFL that clients do not receive backward gradients at the split boundary, and our hybrid ZO–FO design is built **directly on** this structure. Accordingly, we **do not claim** that eliminating activation caching is a generic advantage over FL/SL. Rather, given an SFL system, we show that client-side FO can be safely replaced by ZO at the split layer such that (i) training performance on the same architecture and task is preserved, and (ii) client-side memory and computation are significantly reduced. This is why our comparisons focus on FLOPs and memory metrics against other SFL methods under a fixed SFL setting, instead of conducting a four-way comparison with FL/SL, which operate under different training regimes where clients are assumed capable of full-model training.
> > >
> > > 2. Our work is an algorithmic contribution within the standard synchronous SFL regime, where SFL is **not interchangeable with FL or SL**. SFL specifically targets settings where clients cannot train the full model locally due to memory/compute limits, which makes direct FL/SL comparisons inherently unfair. Moreover, switching to FL or SL does not guarantee lower latency: FL requires full-model backpropagation on every client, while SL incurs additional forward/backward synchronization. Under the SFL protocol, **HERON-SFL introduces no extra latency**, as it uses exactly the same communication pattern as prior auxiliary-network SFL methods (transmitting smashed activations and auxiliary-network gradients only). Following standard SFL practice, we therefore report accuracy per communication round and show that HERON-SFL matches or surpasses other SFL baselines under equal global updates. A full wall-clock comparison across FL/SL/SFL would rely heavily on deployment-specific hardware and network assumptions and is outside the scope of an algorithmic SFL study.
> > >
> > > 3. We would like to emphasize that HERON-SFL is not restricted to cross-silo scenarios. The method itself does not make assumptions about client type. In the revised manuscript (Appendix D.1.2), we further added experiments with larger numbers of participating clients, which show that the algorithm scales well as the population grows.
> > >
> > > 4. We emphasize that the low-effective-rank behavior is widely observed in modern deep models under standard training [1–4]. For example, [1] shows that various ResNet models exhibit clear low-rank structure in the later stages of training, and MeZO [3] further demonstrates in LLM fine-tuning that the effectiveness of zeroth-order optimization relies critically on this low–effective-rank property. The special case with strong soft-orthogonality regularization is introduced to deliberately suppress this effect—represents a substantially altered training regime **rather than typical practice**. Our assumption is intended to capture this standard training behavior rather than such heavily regularized cases.
> > >
> > > [1] Behrooz Ghorbani et al., “An investigation into neural net optimization via Hessian eigenvalue density,” ICML 2019.
> > > [2] Jiawei Zhao et al., “Galore: Memory-efficient LLM training by gradient low-rank projection,” ICML 2024.
> > > [3] Malladi et al., “Fine-tuning language models with just forward passes,” NeurIPS 2023.
> > > [4] Zhe Li et al., “Dimension-Free Communication in Federated Learning via Zeroth-Order Optimization,” ICLR 2025.

---

### Official Review · Reviewer_n2hD · 2025-10-31

**Soundness:** 2
**Presentation:** 3
**Contribution:** 2
**Rating:** 2
**Confidence:** 4

**Summary:**

This paper proposes a method for reducing computation and communication overhead in split learning while preserving model accuracy. The key idea is to enable clients to train local models with new auxiliary components so that the clients can keep doing the local training without waiting for the backpropagation results from the server. Server does receive the smashed activations from clients, but not for every forward pass in clients.

**Strengths:**

The paper targets a highly relevant challenge of how to scale federated training to resource-limited clients without sacrificing global model quality. The proposed method preserves split learning’s efficiency benefits (clients train partial models) while avoiding its synchronization and overheads. The experiments are broad.

**Weaknesses:**

1. The idea is very similar to FedGKT [1], which has not been compared with. There are many works that are follow ups of FedGKT, the authors need to cover some of the recent ones in their comparisons.
2. The smashed activations can leak data privacy. It has to be experimentally demonstrated how the proposed scheme is robust to model inversion and other attacks.
3. While not directly FL, there is another recent work that seeks to train small models at clients with support from server by offloading some intermediate activations with guaranteed DP privacy [2]. It seems the current paper is an extension of [2]. Instead of auxiliary network, [2] directly creates a smaller model for the user. I think that is a more principled approach for extension to FL.

[1] Group Knowledge Transfer: Federated Learning of Large CNNs at the Edge (NIPS 2020).
[2] All Rivers Run to the Sea: Private Learning with Asymmetric Flows (CVPR 2024).

**Questions:**

Please addresses the weaknesses

---

> ### Author Response · Authors · 2025-11-21
>
> Thank you for your comments and suggestions.
> > Concern 1:  Similarity to FedGKT and CVPR 2024 ‘All Rivers Run to the Sea’ (Weakness 1, 3)
>
> Thank you for raising this point.
> * FedGKT belongs to the family of knowledge-distillation-based FL, where clients train small auxiliary models and transfer logits to the server, which uses a distillation loss to supervise its local model. As discussed in FSL-SAGE [3], **FedGKT is not an SFL method**, because it does not partition a single large model across client and server and does not maintain consistency of forward/backward paths. Its training objective is fundamentally different from SFL methods, and its performance is not directly comparable to our HERON-SFL.
> * We also note that [2] tackles a completely different problem: it is a privacy-preserving training/inference framework that relies on asymmetric IR decomposition (SVD+DCT) and DP-protected residual offloading between a TEE and a public GPU. In contrast, HERON-SFL operates in a multi-client Split Federated Learning setting, where the split architecture and communication pattern are fixed, and the contribution lies purely in improving the optimizer (replacing FO with ZO). Thus, Delta is a system-architecture redesign, whereas HERON-SFL is an optimization-level improvement within the standard SFL pipeline. For this reason, the two methods target orthogonal dimensions and are not directly comparable.
>
> [1] Group Knowledge Transfer: Federated Learning of Large CNNs at the Edge (NeurIPS 2020).
> [2] All Rivers Run to the Sea: Private Learning with Asymmetric Flows (CVPR 2024).
> [3] FSL-SAGE: Accelerating Federated Split Learning via Smashed Activation Gradient Estimation (ICML 2025).
>
> > Concern 2: Privacy Leakage
>
> We acknowledge your concern about the privacy risks of sending cut-layer activations to the server. However, such risks are inherent to any SL method that transmits smashed data. A detailed analysis of attacks and privacy guarantees is important but beyond the scope of this work. We also note that our SFL setting is the same as in other SFL algorithms published in the literature, hence having the same privacy performance.

---

### Official Review · Reviewer_EnDt · 2025-11-01

**Soundness:** 2
**Presentation:** 2
**Contribution:** 1
**Rating:** 2
**Confidence:** 4

**Summary:**

The paper introduces HERON-SFL, a Split Federated Learning framework that uses zeroth-order (ZO) optimization for client-side updates and first-order (FO) optimization on the server side. This hybrid approach reduces client-side memory and computational costs by eliminating backpropagation and activation caching. Empirical results show significant reductions in client memory and computational cost while maintaining comparable accuracy.

**Strengths:**

Using ZO optimization on the client side show significant resource savings (up to 64% memory and 65% computation) with comparable accuracy, making it suitable for resource-constrained devices.

**Weaknesses:**

1.	The main contribution of HERON-SFL is an incremental update to Han et al.'s local-loss-based split learning. In Han et al., clients perform local updates using their own auxiliary model eliminating global backprop. HERON-SFL simply swaps the FO updates on the client-side with ZO updates while keeping the server-side optimization FO-based.
2.	The core idea, that ZO optimization can replace FO optimization for memory reduction, is not new, as ZO optimization has been explored extensively before.
3.	While ZO optimization can reduce memory usage, it introduces several new hyperparameters (perturbation size µ, number of probes etc), which can make tuning more complex. The sensitivity of ZO to these hyperparameters is a critical issue which is not addressed or ablated in this paper.
4.	ZO optimization is typically slow and suffers from higher variance which suggest client-side may face unstable updates. While the FO server-side optimization may mitigate this to some extent in the current experiments, the instability inherent to ZO methods is a known issue which is not clarified.
5.	The paper provides theoretical convergence guarantees, but relies on low effective rank assumption, which may not always hold, especially for high-dimensional models.
6.	The empirical experiments are limited to ResNet-18 (CIFAR-10) and GPT-2 fine-tuning with relatively small numbers of clients (N=5 for CIFAR-10 and N=3 for GPT-2). These do not demonstrate the scalability of HERON-SFL to larger models, larger datasets and more clients.

**Questions:**

1.	Could you clarify how many ZO probes per mini-batch/step were used in the experiments, and how the communication cost was accounted for when these probes are performed?
2.	In Table 2, the peak memory claims for HERON-SFL seem to suggest an O(1) memory model—can you provide a more detailed breakdown of how memory is measured (e.g., parameters, activations, optimizer states)? Can you compared this cost to Han et. al.’s method?
3.	The resource cost analysis in Table 1 does not correspond well with the consumptions reported in Table 2 and Table 3. The flops count of HERON-SFL is expected to be 2/3rd to that of FSL_SAGE but is 3 times less in Table 2. Can you discuss why this discrepancy exists?

---

> ### Author Response · Authors · 2025-11-21
>
> Thank you for your comments and suggestions.
> > Weakness 1: Mariginal Contribution; Incremental ZO drop-in for clients;
>
> Thank you for examining our method’s relationship with [1]. We would like to clarify that **HERON-SFL does not merely replace client-side FO with ZO**.  In [1], the core contribution is the introduction of local loss into split federated learning, enabling clients to perform auxiliary-model-based local updates without requiring global backpropagation. Our work addresses a fundamentally different design objective: improving the update capability of resource-constrained clients by removing their backward-pass memory and computation bottlenecks. From this perspective, the main contribution of HERON-SFL is not to modify [1]’s local-loss idea, but to develop a **hybrid FO–ZO optimization scheme** that makes local updates feasible on devices that cannot afford first-order backprop. The motivation, constraints, and resulting client-side resource profile of HERON-SFL differ substantially from [1]’s formulation.
>
> [1] Han, Dong-Jun et al. "Accelerating Federated Learning with Split Learning on Locally Generated Losses." (2021).
>
> >  Weakness 2: ZO-based memory reduction is not new;
>
> ZO methods have indeed been studied elsewhere, but their role in split federated learning is fundamentally different. Prior ZO work assumes centralized or full-model FL training and **does not address SFL-specific constraints** such as cross-device backpropagation and the large activation memory footprint. In HERON-SFL, ZO is applied only to the client partition and must interact with auxiliary networks to enable a fully forward-only local pipeline—an SFL design problem that has not been explored before. Moreover, our convergence result is tied to the ZO–FO hybrid structure under low effective rank and is not implied by existing ZO analyses. These differences show that HERON-SFL is more than simply applying an existing optimizer.
>
>
> > Weakness 3 & 4: ZO sensitivity and ablation study; Instability of ZO;
>
> * While ZO introduces hyperparameters such as the perturbation scale $\mu$ and probe count, in practice, these parameters are not sensitive in the SFL setting. We include an ablation in the revision (see Appendix D.2) showing that varying $\mu$ and the number of probes across standard ranges leads to negligible accuracy changes, confirming that tuning complexity is very limited in our framework in resource-constrained SFL settings.
> * The instability often associated with pure ZO methods does not arise in HERON-SFL. This is not due to any refinement from the server-side optimizer, but because the client-side ZO operates on a shallow, low-rank-effective partition whose loss landscape naturally supports stable and low-variance optimization. The server-side FO is used only to train the global model through standard backpropagation, ensuring conventional convergence behavior without interacting with or modifying the ZO updates.

---

> > ### Author Response · Authors · 2025-11-21
> >
> > > Weakness 5: low effective-rank assumption;
> > Please see our response to `Weakness 4 by FesG`.
> >
> >
> > > Weakness 6: Scalability;
> >
> > * The main contribution of this work is to improve the efficiency of client-side training in resource-constrained SFL systems. Accordingly, we evaluate HERON-SFL in two complementary regimes that represent the dominant use cases of SFL: (i) training a vision model from scratch (ResNet-18 on CIFAR-10), and (ii) fine-tuning a substantially larger language model (GPT-2) on E2E. Although the tasks themselves differ, they stress the SFL pipeline along different axes (full-model training versus parameter-efficient finetuning) and together demonstrate that HERON-SFL remains effective across both lightweight and substantially heavier client–server model splits.
> > * Regarding scalability with respect to the number of clients, we have included additional experiments in the revised version (see Appendix D.1.2) demonstrating that HERON-SFL **maintains stable performance** when the number of participating devices increases, confirming that the proposed method extends naturally to larger-client SFL deployments.
> >
> > > Question 1: ZO probes per mini-batch/step; communication cost;
> >
> > * We use two-point ZO probes per optimization step (two forward passes) in experiments.
> > * Only the clean smashed activations are sent to the server; perturbed activations stay local. Therefore, the communication cost is identical to standard auxiliary-network SFL and independent of the number of ZO probes.
> >
> > > Question 2: Memory Measurement; Compare with [1];
> >
> > Client memory in HERON-SFL consists only of parameters, one mini-batch of activations, and optimizer states. Because ZO requires no backward pass, no intermediate activations are cached, yielding $\mathcal O(1)$ activation memory. We add details in the revision (see Appendix C.1)
> >
> > Han et al.’s method [1] has the same client-side peak memory as CSE-FSL [2], since both still perform local FO backprop. The main difference is that [1] requires multiple distributed server devices, which introduces substantial server-side communication and storage overhead. Because prior work [2] has established clear communication-efficiency gains over Han et al. [1], benchmarking against [2] alone is adequate.
> >
> > [1] Han, Dong-Jun et al. "Accelerating Federated Learning with Split Learning on Locally Generated Losses." (2021).
> > [2] Mu et al. "Communication and storage efficient federated split learning." ICC 2023-IEEE International Conference on Communications. IEEE, 2023.
> >
> > >Question 3: resource cost analysis;
> >
> > Thank you for raising this issue. We realized that in Table 2, we only accounted for the FLOPs of a single perturbation. The per-step cost should indeed include two identical perturbations, resulting in 39.9G FLOPs. We have updated the results and the corresponding analysis (as well as discussion in the abstract and introduction) in the revision. We appreciate the reviewer’s careful observation, which helped us improve the completeness and accuracy of our work.

---

### Note · Authors · 2025-12-22

I have read and agree with the venue's withdrawal policy on behalf of myself and my co-authors.